**Seasonal cycles of biogeochemical fluxes in the Scotia Sea, Southern Ocean: A stable isotope**
**approach**
Anna Belcher[1], Sian F. Henley[2], Katharine Hendry[1,3], Marianne Wootton[4], Lisa Friberg[3], Ursula
Dallman[2], Tong Wang[3], Christopher Coath[3], Clara Manno[1]
[1] British Antarctic Survey, Cambridge, CB3 0ET, UK
[2] School of GeoSciences, University of Edinburgh, Edinburgh EH9 3FE, UK
[3] University of Bristol, Bristol, BS8 1RJ, UK
[4] Marine Biological Association, Plymouth, PL1 2PB, UK
Correspondence to: Anna Belcher (annbel@bas.ac.uk) and Clara Manno (clanno@bas.ac.uk)
**Abstract**
The biological carbon pump is responsible for much of the decadal variability in the ocean carbon
dioxide ($CO_2$) sink, driving the transfer of carbon from the atmosphere to the deep ocean. A
mechanistic understanding of the ecological drivers of particulate organic carbon (POC) flux is key to
both the assessment of the magnitude of the ocean $CO_2$ sink, as well as for accurate predictions as to
how this will change with changing climate. This is particularly important in the Southern Ocean, a
key region for the uptake of $CO_2$ and the supply of nutrients to the global thermocline. In this study
we examine sediment trap derived particle fluxes and stable isotope signatures of carbon (C),
nitrogen (N) and biogenic silica (BSi) at a study site in the biologically productive waters of the
northern Scotia Sea in the Southern Ocean. Both deep (2000 m) and shallow (400 m) sediment traps
exhibited two main peaks in POC, particulate N and BSi flux, one in austral spring and one in
summer, reflecting periods of high surface productivity. Particulate fluxes and isotopic compositions
were similar in both deep and shallow sediment traps, highlighting that most remineralisation
occurred in the upper 400 m of the water column. Differences in the seasonal cycles of isotopic
compositions of C, N and Si provide insights into the degree of coupling of these key nutrients. We
measured increasing isotopic enrichment of POC and BSi in spring, consistent with fractionation
during biological uptake. Since we observed isotopically light particulate material in the traps in
summer, we suggest physically-mediated replenishment of lighter isotopes of key nutrients from
depth, enabling full expression of the isotopic fractionation associated with biological uptake. The
change in the nutrient and remineralisation regimes, indicated by the different isotopic
compositions of the spring and summer productive periods, suggests a change in the source region
of material reaching the traps and associated shifts in phytoplankton community structure. This,
combined with the occurrence of advective inputs at certain times of the year, highlights the need to
make synchronous measurements of physical processes to improve our ability to track changes in
the source regions of sinking particulate material. We also highlight the need to conduct particle-
specific (e.g. faecal pellet, phytoplankton detritus, zooplankton moults) isotopic analysis to improve
the use of this tool in assessing particle composition of the sinking material and to develop our
understanding of the drivers of biogeochemical fluxes.

## 1. Introduction

The transfer of carbon from the atmosphere to the deep ocean via the biological carbon pump (Volk
and Hoffert, 1985) is important for the sequestration of carbon, and combined with ocean
circulation is a main driver of decadal variability of the ocean carbon dioxide ($CO_2$) sink (DeVries,
2022). Mechanistic understanding of the processes controlling the magnitude and efficiency of the
biological carbon pump is therefore key to assessment and prediction of the ocean's role as a $CO_2$
sink and requires robust characterisation of the composition of the sinking particles transferring
particulate organic carbon (POC) to the deep ocean. The composition of particles affects the sinking
rate, lability and thus degree of remineralisation as they sink through the water column (e.g. Ploug
et al., 2008; Giering et al., 2020).
Sediment traps enable visual assessment of sinking particles, and have been deployed in numerous
locations throughout the world's oceans to both quantify biogeochemical fluxes and characterise the
nature of sinking material (e.g. Torres Valdés et al., 2014). Sediment traps can be susceptible to
collection biases depending on the depth of deployment, trap design, hydrodynamic conditions and
properties of sinking particles (Buesseler et al., 2007). Moored sediment traps can underestimate
the actual flux at depths shallower than ~1500 m by collecting only a portion of the sinking material,
though biases vary greatly between sites (Buesseler et al., 2007). Numerous studies have recorded
the dominance of particular organisms or types of detrital material in trap material, highlighting the
importance of ecosystem community structure on the magnitude and efficiency of the biological
carbon pump. For example, faecal pellets, diatoms, diatom resting spores and acantharia have been
observed as significant contributors to particle fluxes (González et al., 2009; Belcher et al., 2018,
2017; Manno et al., 2015; Gleiber et al., 2012; Rembauville et al., 2015; Roca-Marti et al., 2017).
Such visual assessment of trap material is typically very time consuming. Additionally, fragile
material, such as salp faecal pellets (Iversen et al., 2017; Pauli et al., 2021) may break up in the
sample manipulation processes, making them hard to account for visually. Biogeochemical methods
such as the use of stable isotopes may offer additional insight into the drivers of POC fluxes (e.g.
Henley et al., 2012).
Marine phytoplankton take up aqueous $CO_2$ ($CO_{2(aq)}$) during photosynthesis, converting it to organic
carbon. During this process, the lighter isotope ($^{12}C$) is preferentially assimilated, which enriches the
residual aqueous pool in the heavier isotope ($^{13}C$). The stable isotopic composition of the POC
($\delta^{13}C_{POC}$) of the marine phytoplankton is therefore lower than that of the carbon source. Over large
scales, the $\delta^{13}C$ of marine phytoplankton has been found to be inversely correlated with [$CO_{2(aq)}$] in
surface waters (Rau et al., 1991). However, numerous other factors have been identified as
impacting the $\delta^{13}C_{POC}$ of surface waters and marine plankton. Phytoplankton growth rates, cell
geometry and non-diffusive uptake of carbon via carbon concentration mechanisms have all been
highlighted as impacting the $\delta^{13}C_{POC}$ of marine plankton and thus surface waters (Popp et al., 1999,
1998; Bidigare et al., 1999; Trull and Armand, 2001; Tuerena et al., 2019). This decoupling of the
relationship between $\delta^{13}C_{POC}$ and [$CO_{2(aq)}$] presents challenges for palaeoceanographic studies, but
also the possibility of using the $\delta^{13}C_{POC}$ of marine samples to infer information about community
composition.
During photosynthetic uptake, the balance between supply and demand of carbon impacts $\delta^{13}C_{POC}$,
regulated by the transport of inorganic carbon into the internal cell and fixation to organic carbon
(Popp et al., 1999; Trull and Armand, 2001). A greater isotopic fractionation occurs in smaller
phytoplankton cells, enabled by the higher cell surface area to volume (SA:V) ratios and increased
amount of $[CO_{2(aq)}]$ diffusing across the cell membrane relative to the total carbon within the cell
(Popp et al., 1998; Tuerena et al., 2019; Hansman and Sessions, 2016). Thus, a community
dominated by large, fast-growing diatoms is expected to contribute to enriched $\delta^{13}C_{POC}$ values
compared to a community dominated by picoplankton. A study by Henley et al. (2012) in the coastal
western Antarctic Peninsula, attributed a large (~10 ‰) negative isotopic shift in $\delta^{13}C_{POC}$ to a near-
complete biomass dominance of the marine diatom *Proboscia inermis,* highlighting the possible
impact of shifts in species composition on stable isotopes. It may therefore be possible to use stable
isotopes to gain information about the community composition of phytoplankton driving, for
example, large spring pulses in POC flux. Additionally, siliceous phytoplankton, such as diatoms,
require dissolved silica (silicic acid, or DSi) to build their cell walls or frustules (amorphous $SiO_2.nH_2O$,
referred to here as biogenic silica, BSi). During uptake of DSi, diatoms fractionate the stable isotopes
of silicon ($^{28}Si$, $^{29}Si$, $^{30}Si$), preferentially taking up the lighter isotopes during cell wall (frustule)
formation (De La Rocha et al., 1997). This means that BSi fluxes and ratios of light $^{28}Si$ to heavy $^{30}Si$
(expressed as $\delta^{30}Si$) in sinking particulate organic matter (POM) can be informative about DSi
utilisation by siliceous phytoplankton. The fractionation of Si isotopes during diatom DSi utilisation is
approximately -1.1 ‰, although estimates of this value vary in laboratory and field studies between
-0.5 and -2. 5 ‰ (Hendry and Brzezinski, 2014). Whilst some studies have shown that isotopic
fractionation is independent of temperature, DSi concentration and diatom species (e.g., De La
Rocha et al., 1997), one *in vitro* laboratory culture experiment revealed a potential species effect,
with polar species exhibiting more extreme fractionation (-2.09 ‰ for *Chaetoceros* sp. and 0.54 ‰
for *Fragilariopsis kerguelensis*; Sutton et al., 2013). The impact of water column dissolution on
frustule $\delta^{30}Si$ is poorly constrained, with experimental evidence for either a small fractionation of -
0.55 ‰ (Demarest et al., 2009) or a negligible impact (Wetzel et al., 2014; Egan et al., 2012; Grasse
et al., 2021).
Additionally, the stable isotopes of marine nitrogen reveal information about uptake of inorganic
nitrogen sources by phytoplankton (Wada and Hattori, 1978), as well as trophic and food web
processes (Michener and Lajtha, 2008). Nitrogen has two isotopes, $^{14}N$ and $^{15}N$, and the ratio
between these heavy and light isotopes is expressed as $\delta^{15}N$. Different sources of nitrogen can alter
the stable isotopic composition of marine phytoplankton because ammonium characteristically has a
lower value of $\delta^{15}N$ than nitrate supplied from depth. As well as this, isotopic fractionation occurs
during transfer through the food web, with a trophic enrichment of typically 2- 4 ‰ between
successive trophic levels (Montoya, 2007; Minagawa and Wada, 1984). Excretion and egestion
processes can also impact $\delta^{15}N$; isotopic discrimination during excretion of ammonium by
zooplankton and fish results in ammonium that is $^{15}N$-depleted relative to the substrate catabolised
(Montoya, 2007). Thus, there are several interacting processes impacting the degree of fractionation
and subsequent isotopic ratios in particulate nitrogen (PN) and knowledge of $\delta^{15}N$ ratios may
provide insight into biogeochemical processes and the composition of the sinking flux.
In this study we examine the seasonal cycle of the magnitude and composition of vertical
biogeochemical fluxes of particulate material collected by two sediment traps deployed for almost
one year on a deep ocean mooring located in the northern Scotia Sea in the Atlantic sector of the
Southern Ocean. The Scotia Sea, particularly the region downstream of South Georgia, is a hot spot
for biological productivity, supported by higher iron availability (Korb et al., 2008; Matano et al.,
2020). Diatoms dominate the phytoplankton assemblage, particularly in the summer months, with
smaller contributions of dinoflagellates (Korb et al., 2012). The large, consistent phytoplankton
blooms occurring in this region support high fluxes of POC to the deep ocean, with two peaks in POC
flux occurring during the seasonal cycle; the first peak in austral spring, and the second in late
summer or early autumn (Manno et al., 2015). Faecal pellets (up to 91 % in late spring and early
summer; Manno et al., 2015), krill exuviae (up to 47 % in summer; Manno et al., 2020) and diatoms,
particularly resting spores (annual contribution of 42 %; Rembauville et al., 2016) have been shown
to make large contributions to the POC fluxes in our study region. Here we use $\delta^{13}C_{POC}$, $\delta^{15}N_{PN}$ and
$\delta^{30}Si_{BSi}$ alongside calculated fluxes of POC, PN and BSi as tools to reveal information about sinking
particulate organic matter and the processes influencing its production and subsequent flux to
depth. More in-depth understanding of the composition, and thus the drivers of POC flux in this
important region are key to improving estimates of the current and future strength of the biological
carbon pump and the ocean's role as a $CO_2$ sink.

**2. Methods**
*2.1. Study Area*
This study was conducted in the open ocean environment of the northern Scotia Sea in the Southern
Ocean at a long-term observatory station, P3 (Figure 1), where an oceanographic mooring is located.
The mooring is part of the Scotia Sea Open Ocean Observatory (SCOOBIES:
https://www.bas.ac.uk/project/scoobies/), a programme designed to investigate the biological and
biogeochemical influence of the large and persistent phytoplankton bloom to the northwest of
South Georgia.

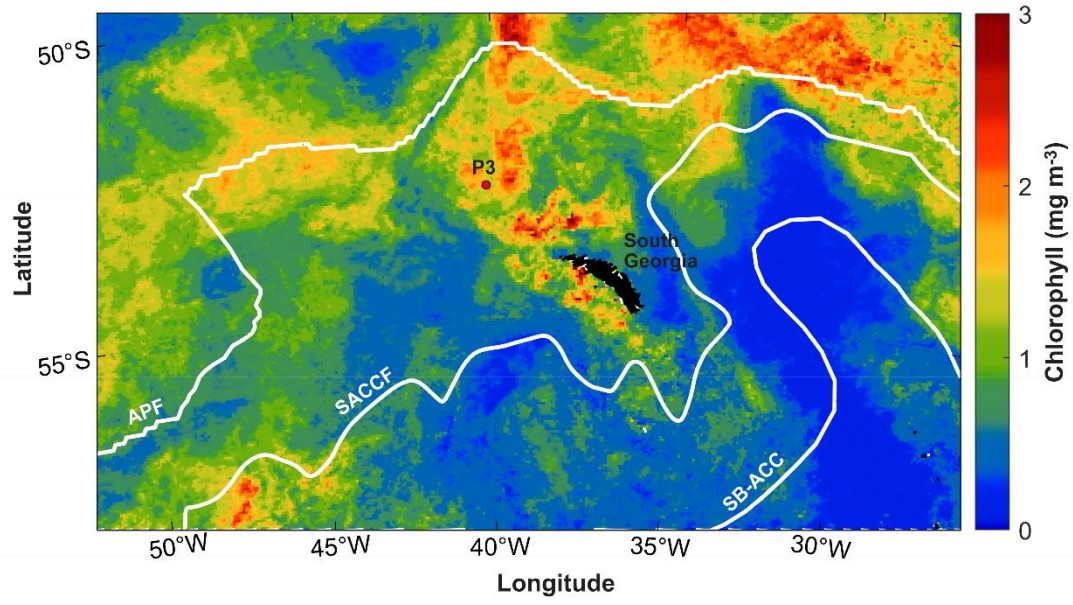


 Figure 1: Location of P3 mooring site to the northwest of South Georgia. White lines indicate

frontal positions of the Antarctic Polar Front (APF) (Moore et al., 1999), Southern Antarctic

Circumpolar Current Front (SACCF) (Thorpe et al., 2002) and the Southern Boundary of the

Antarctic Circumpolar Current (SB-ACC) (Orsi et al., 1995). Mean chlorophyll concentration (mg m$^{-3}$)

is shown for December 2018 from 8-day satellite chlorophyll data from the Ocean Colour CCI

(version 5.0) (Sathyendranath et al., 2021, 2019).

### 2.2. Sediment trap deployment

Two sediment traps were deployed on the mooring array to collect sinking particles for analysis of

carbon, nitrogen and biogenic silica fluxes and analysis of $\delta^{13}C_{POC}$, $\delta^{15}N_{PN}$ and $\delta^{30}Si_{BSi}$. The mooring

was deployed from 25$^{th}$ January 2018, during research cruise JR17002 aboard the *RRS James Clark

Ross,* to 1$^{st}$ January 2019, recovered during research cruise DY098 aboard the *RRS Discovery.* The

mooring was located at 52.8036 °S, 40.1593 °W, to the northwest of South Georgia island in the

Scotia Sea at a water depth of 3748 m. Sediment traps (McLane PARFLUX, 0.5 m$^2$ surface collecting

area; McLane Research Laboratories Inc, Falmouth, MA, USA) were deployed at 400 and 2000 m

(referred to hereafter as shallow and deep respectively) and were each equipped with 21 sample

bottles. A baffle at the top of the trap prevents large organisms from entering and each sample

bottle contained a formosaline solution (filtered seawater containing 2 % v/v formalin, mixed with

sodium tetraborate (BORAX; 0.025 % w/v), and 0.5% w/v sodium chloride) to prevent mixing with

the overlying water column and stop biological degradation. Previous studies have reported the

effects of formalin on $\delta^{13}C_{POC}$ and $\delta^{15}N_{PN}$ to be small (±1 ‰ and ±1.5 ‰ respectively; Mincks et al.,

2008 and references therein). This equates to 13 % and 16 % of the maximum range measured in our

study, which is small compared to the isotopic shifts we observed. Yet we stress that all $\delta^{13}C_{POC}$ and

$\delta^{15}N_{PN}$ values given here are associated with this uncertainty. The sediment trap sample carrousel

was programmed to rotate every 7-31 days depending on the season; shorter periods to coincide

with austral summer and longer periods during austral winter (Table S1). TM Seaguard current

meters were deployed ~50 m above the shallow sediment trap and 50 m below the deep sediment

trap, set at a measurement interval of 2 hours.

### 2.3. Trap sample processing

Each sample bottle from the sediment trap was processed on return to the laboratory. The

supernatant was carefully removed using a syringe and swimmers (zooplankton that are believed to

have entered the trap actively whilst alive) were removed. Swimmers were removed by hand under

a dissecting microscope and were not included in flux calculations. The material from each sediment

trap sample bottle was split into a number of smaller aliquots for subsequent analysis using a

McLane rotary splitter.

### 2.3.1. Organic carbon and nitrogen

For each sediment trap bottle from both deep and shallow traps, two or three splits were taken and

each analysed for POC and PN mass and $\delta^{13}C_{POC}$ and $\delta^{15}N_{PN}$. Once split, the material was filtered onto

pre-combusted (450 °C, 16h) 25 mm glass fibre filters (GF/F; nominal pore size 0.7 µm) and rinsed

with milli-Q water. Samples were air dried, fumed for 24 h with 37 % HCl in a desiccator, before

finally oven-drying at 50 °C for 24 h. Filters and filter blanks were placed in sterile tin capsules and
POC and PN were measured on a CE Instruments NA2500 elemental analyser, calibrated using an
acetanilide calibration standard with a known %C and %N of 71.09 % and 10.36 % respectively.
Standards were interspersed regularly between samples to measure and correct for drift. Analytical
precision was better than 1.0 % for POC and 1.1 % for PN. The POC flux ($F, mg\ C\ m^{-2}\ d^{-1}$) for each
sample was calculated using the following equation:
$F = m/(A \times d)$                                                    (1)
Here $m$ is the mass of POC in the sample bottle (mg), $d$ is the number of days that the sample bottle
was open (7–31 days) and $A$ is the surface area of the sediment trap opening ($0.5\ m^2$). The same
calculation was carried out for PN.
$\delta^{13}C_{POC}$ and $\delta^{15}N_{PN}$ were analysed on a Thermo Finnigan Delta-V Advantage isotope ratio mass
spectrometer that was in line with the elemental analyser. All $\delta^{13}C_{POC}$ and $\delta^{15}N_{PN}$ data are presented
in the delta per mille (‰) notation relative to the appropriate international standard, according to
equation 2.
$\delta X(‰) = 10^3 \left( R_{sample}/R_{standard} - 1 \right)$                            (2)
$R$ denotes the $^{13}C/^{12}C$ ratio for carbon or the $^{15}N/^{14}N$ ratio for nitrogen. $R_{sample}$ refers to the relevant
ratio in the sample. $R_{standard}$ refers to the ratios in the international standards Vienna Pee Dee
belemnite (V-PDB) for $\delta^{13}C$ and atmospheric nitrogen (AIR) for $\delta^{15}N$ , both of which are calibrated
against the PACS-2 marine sediment reference material. Multiple repeats of analytical standards
gives a reproducibility of 0.2 ‰ for C and N, which is significantly smaller than the uncertainty
associated with organic molecules in the formalin preservative (±1 ‰ and ±1.5 ‰ for C and N
respectively; Mincks et al., 2008 and references therein).

*2.3.2. Biogenic silica*
Two splits were taken from each sample bottle from both deep and shallow sediment traps for
analysis of biogenic silica and silicon isotopes. Split material was filtered onto 25 mm, 0.4 μm,
polycarbonate filters and rinsed with Milli-Q water before drying at 50 °C for 24h. Material on the
filters was solubilised via an alkaline extraction method (Hatton et al., 2019) carried out at the Bristol
Isotope Group (BIG) laboratory. Sample material was digested in Teflon tubes with 0.2M NaOH at
100 °C for 40 minutes. This was followed by neutralisation with 6M HCl. Biogenic silica ($SiO_2$, termed
BSi) concentrations were measured chlorometrically by molybdate blue spectrophotometry
(Heteropoly Blue Method) (Strickland and Parsons, 1972) using a Hach DR3900 spectrophotometer
set at a wavelength of 815 nm. Supernatants were stored for 7-11 months before column chemistry
for isotope analysis. Fluxes of biogenic silica were calculated as for POC using equation 1.
For Si isotope analysis, supernatants and reference materials were purified by passing through
cation exchange columns (Bio-Rad AG50W-X12, 200-400 mesh resin) pre-cleaned with HCl following
Georg et al. (2006). Samples were acidified to a pH of 1-2 to ensure that all the silicon remained in
solution. Samples were loaded onto columns and eluted with Milli-Q water to produce a 2.5 ppm
solution, and concentrations were checked to confirm quantitative yields. Si isotopic composition
was analysed within 24 hours of column chemistry. Stable Si isotopic compositions are presented in
standard delta notation ($\delta^{30}$Si), as for $\delta^{13}C_{POC}$ and $\delta^{15}N_{PN}$ according to Equation 2, where R is $^{30}$Si/$^{28}$Si.
These compositions are checked against $\delta^{29}$Si (where R is $^{29}$Si/$^{28}$Si) for mass dependence. The
samples were measured at the BIG laboratory on a Finnigan Neptune Plus High-Resolution MC-ICP-
MS (Thermo Fisher Scientific). The Si solutions were spiked with magnesium spike (Inorganic
Ventures MSMG-10 ppm), hydrochloric acid (1M HCl in-house distilled) and sulphuric acid (0.1M
$H_2SO_4$, ROMIL-UpA™ Ultra Purity Sulphuric Acid), and transferred from the autosampler via a PFA
Savillex C-Flow nebulizer (35 µl min$^{-1}$) connected to an Apex IR Desolvating Nebulizer (Ward et al.,
2022), and measured on the low-mass side to resolve any isobaric interferences (e.g., $^{14}N^{16}O^{+}$). All
standards and samples were blank-corrected offline. The intensity of $^{28}$Si in the 0.1M HCl blank was
<1 % of the sample intensity in all sample runs. Furthermore, we also measured Mg isotopes ($^{24}$Mg,
$^{25}$Mg and $^{26}$Mg) as an internal isotopic reference to correct for any mass-dependent fractionation
(Cardinal et al., 2003). Measurements that resulted in large corrections (>0.3 ‰ on $\delta^{30}$Si) underwent
repeat analysis. Instrumental mass bias was further accounted for using a standard-sample
bracketing method using a 2 ppm reference standard (NBS or RM8546) solution. Two splits were
analysed for each sediment trap bottle, as well as standards and sample blanks. Solutions obtained
from each split were measured in replicate (n = 2-3) alongside continuous measurement of
reference materials Diatomite and LMG-08 to ensure reproducibility and to monitor data quality.
Measurements of Diatomite and LMG-08 yielded $\delta^{30}$Si of +1.23 ‰ (SD ± 0.03, n=18) and -3.40 ‰ (SD
± 0.05, n=5) respectively, which agreed with published values (Reynolds et al., 2007; Hendry and
Robinson, 2012; Grasse et al., 2017). Typical reproducibility between the sediment trap sample splits
(coming from the same sediment trap bottle) was 0.034 ‰ (1 x SD). A lithogenic correction
(e.g.,Closset et al., 2015) was not carried out on these samples given the high percentage of biogenic
silica present in the samples (mean percentage BSi as $SiO_2$ of 17 %). BSi extraction methods show
lower variability for marine sediments with BSi > 15-20 % and do not show evidence for significant
leaching of lithogenic material through time (Conley, 1998). However, even an extreme scenario of
variable lithogenic contamination of 1-5 % of isotopically light marine clays (with $\delta^{30}$Si of -2.3 ‰;
Opfergelt and Delmelle, 2012) would only result in a potential systematic offset of 0.12 ‰, which,
although this is larger than the uncertainty on an individual datapoint, is an order of magnitude
smaller than the observed seasonal signal.
*2.4. Chlorophyll and phytoplankton community composition*
Surface chlorophyll concentrations were obtained from satellite-derived 8-day Ocean Colour CCI
(version 5.0) (Sathyendranath et al., 2021, 2019). We present the monthly mean of these 8-day data
for December at our study site (Figure 1), as well as the 8-day chlorophyll concentration data from
September 2017 to December 2018 (Figure 2) averaged over a 1 x 1° bounding box around our study
site (41 °W, 40 °W, 53 °S, 52 °S).
Light microscopy was used to assess phytoplankton and microzooplankton community composition
of a small selection of samples from the two main productive periods. A biological method of sample
preparation and analysis was chosen, comparable with Rembauville et al. (2015), to determine the
quantity of empty and full cells. Following subsampling using the rotary splitter, samples for
morphological taxonomic analysis were diluted to a standardised 25 ml. Samples were gently
inverted using the Paul Schatz principle (figures of eight) for one minute to homogenise them, and 2
ml was withdrawn using a modified pipette with widened opening. Several common diatoms in
Antarctic waters are long and slim; in particular, *Thalassiothrix antarctica* has been recorded with an

apical axis up to 5mm. To ensure such specimens remain intact and are not excluded from the pipetting process, a wide bore opening was used. The 2 ml subsamples were used to fill a 1 ml Sedgwick Rafter counting chamber. Chambers were viewed using a compound light microscope (Nikon Eclipse 80i) with differential interference contrast at x200 magnification. For the larger, easily identifiable cells, the whole chamber was observed; for smaller cells a proportion of the chamber was examined depending upon cell abundance (at least 500 cells were counted). Only complete cells were enumerated to avoid over counting of fragmented specimens. Cells were determined as "full" or alive at time of collection if they possessed chloroplasts/plastids, pigment, a nucleus or, in the case of *Pronoctiluca*, a distinct accumulation body; cells lacking these internal features were deemed as "empty", or dead at time of collection. Specimens were identified according to Hasle and Syvertsen (1997), Medlin and Priddle (1990), Priddle and Fryxell (1985) and Scott and Marchan (2005).

Cell bio-volume and surface area estimates were calculated using geometrics and the appropriate shape-related equations for phytoplankton genera proposed by Hillebrand et al. (1999). Metrics used in the calculations were based on the average size of ten randomly selected specimens belonging to a species or other taxonomic group within the samples.

## 3. Results

### 3.1. Environmental conditions

Mean current velocities were 0.11 (±0.06) and 0.06 (±0.03) m s$^{-1}$ for shallow and deep current meters respectively (Supplementary Figure S1). Maximum current speeds recorded reached 0.43 and 0.18 m s$^{-1}$ for shallow and deep meters respectively. The periods with currents substantially elevated above the mean were June for both traps, and additionally in late August/September for the shallow trap, both for periods of ~5-10 days. Both are periods of low fluxes during austral winter and are not the main subject of the study here, though it is likely that particle collection was biased at these times (Buesseler et al., 2007).

Satellite-derived estimates of surface chlorophyll show high concentrations during austral summer (January to March) peaking at 2.3 mg m$^{-3}$, as well as during spring (November-December), peaking at 2.1 mg m$^{-3}$ (Figure 2, Figure S2). Data coverage is limited in the winter due to cloud cover, but concentrations appear to be <0.4 mg m$^{-3}$. We define here two productive periods (when chlorophyll concentrations were >0.4 mg m$^{-3}$), which we refer to throughout the manuscript, productive period 1: January to the start of April 2018, and productive period 2: September to the end of December 2018. We note that our sediment trap data begins on the 25$^{th}$ January so we do not capture the start of period 1.

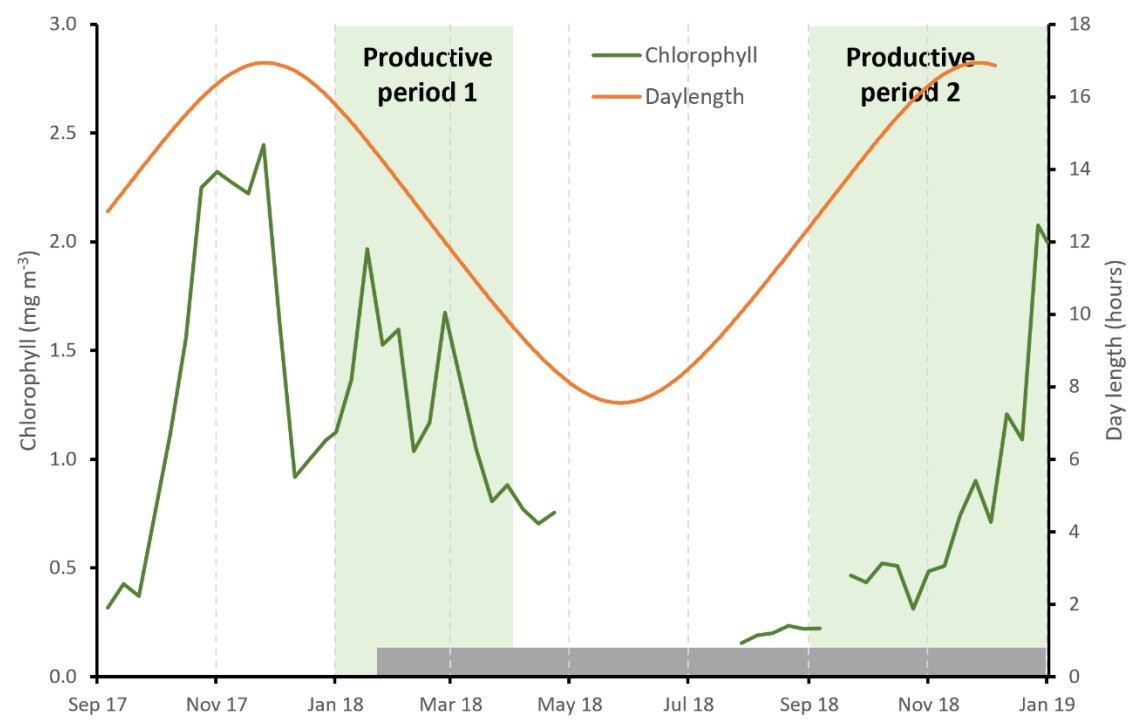


*Figure 2: Seasonal cycle of satellite derived surface chlorophyll concentration (green line, 8-day*
*data from the Ocean Colour CCI (version 5.0)* (Sathyendranath et al., 2021, 2019)*). Daylength at 53*
*°S is shown by the orange line. The two productive periods are highlighted by the shaded green*
*regions, and the grey shaded bar shows the duration of the sediment trap sampling period.*

3.2. POC, PN, BSi fluxes
There is a clear seasonal cycle in POC, PN and BSi fluxes, all tracking each other well (Figure 3). Since
two to three splits were analysed from each sediment trap bottle, we refer here to the mean flux for
each sediment trap bottle based on the available splits for that bottle. POC fluxes were low during
austral autumn and winter, with fluxes <10 mg C m$^{-2}$ d$^{-1}$ and <7 mg C m$^{-2}$ d$^{-1}$ for shallow and deep
traps respectively during the period March to October 2018 (Figure 3A). Higher fluxes were
measured in summer 2018 (productive period 1), reaching 25.3 mg C m$^{-2}$ d$^{-1}$ in late January 2018 in
the shallow trap and 13.1 mg C m$^{-2}$ d$^{-1}$ in late February in the deep trap. The maximum POC fluxes
measured occurred in early December 2018 (productive period 2), reaching 45.7 mg C m$^{-2}$ d$^{-1}$ and
43.4 mg C m$^{-2}$ d$^{-1}$, in shallow and deep traps respectively. PN fluxes follow the same trends as POC
fluxes, peaking at 4.2 and 2.4 mg N m$^{-2}$ d$^{-1}$ during period 1, and 10.8 and 8.2 mg N m$^{-2}$ d$^{-1}$ during
period 2, in shallow and deep traps respectively (Figure 3B). The mean POC:PN ratio (mol:mol)
throughout the study period was 6.40 (± 0.73) and 6.02 (±0.90) in shallow and deep traps
respectively, with higher ratios in the productive periods compared to the winter months. Mean
POC:PN ratios were 6.83 (±0.48) and 6.63 (±0.71) during period 1 and period 2 in the shallow trap,
and 6.40 (±0.63) and 5.51 (±0.87) in the deep trap. Over the winter months POC:PN was 5.83 (±0.54)
and 6.26 (±0.87) in shallow and deep traps respectively.
BSi fluxes (Figure 3C) track those of POC well. Lowest fluxes (<20 mg $SiO_2$ $m^{-2}$ $d^{-1}$) occurred in the
autumn/winter (March-October), with the exception of a small peak of up to 39.7 mg $SiO_2$ $m^{-2}$ $d^{-1}$ in
May 2018. During summer 2018 (productive period 1), BSi fluxes were high, reaching 129.1 mg $SiO_2$
$m^{-2}$ $d^{-1}$ in early February in the shallow trap and 84.3 mg $SiO_2$ $m^{-2}$ $d^{-1}$ in late February in the deep trap.
By far the highest fluxes were observed in spring 2018 (productive period 2), peaking in early
December at 562.4 mg $SiO_2$ $m^{-2}$ $d^{-1}$, and 285.4 mg $SiO_2$ $m^{-2}$ $d^{-1}$ in shallow and deep traps respectively.
The mean BSi:POC ratio (mol:mol) throughout the study period was 29.82 (± 17.80) and 25.86
(±11.72) in shallow and deep traps respectively. Higher BSi:POC ratios were observed in the shallow
trap in period 1 (38.45 ±10.96), and both shallow and deep traps in period 2 (36.94 ±16.32 and 35.70
±12.10 respectively). BSi:POC ratios were lower in the deep trap during period 1 (23.64 ±6.82). The
correspondence in timing of elevated fluxes of POC, PN and BSi fluxes in the shallow and deep traps
in spring (period 2) highlights that sinking rates must be sufficient (>114 m $d^{-1}$) for particles to travel
the 1600 m between the two traps in the 14 day period that those sediment trap cups were open. In
period 1, there was a time lag of 14 to 35 days between the timing of the maximum POC, PN, and BSi
fluxes in the deep and shallow sediment traps. This suggests sinking rates of 46-114 m $d^{-1}$. However,
we stress that this assumes vertical sinking, which as we discuss in Section 4 is not always the case.

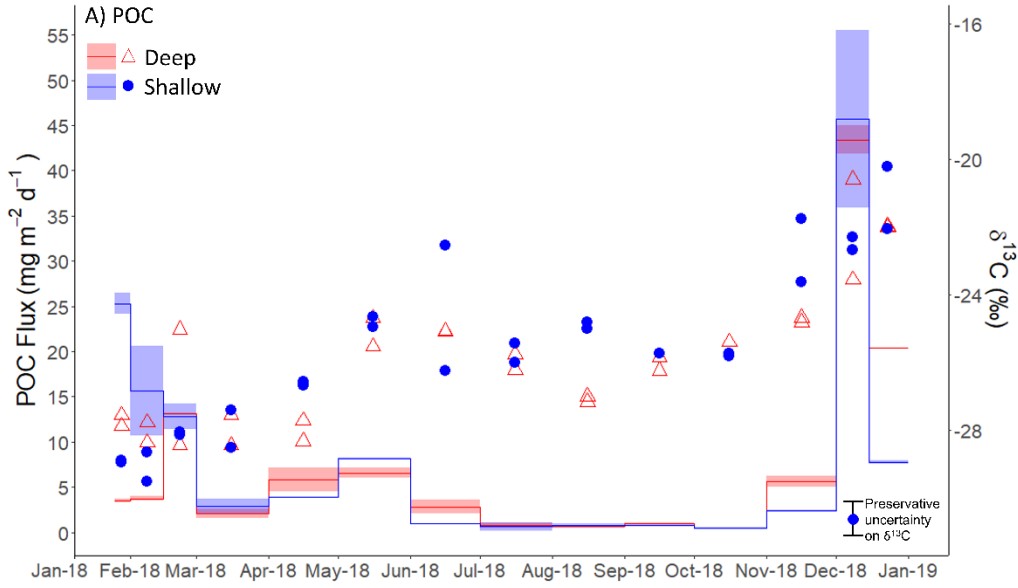

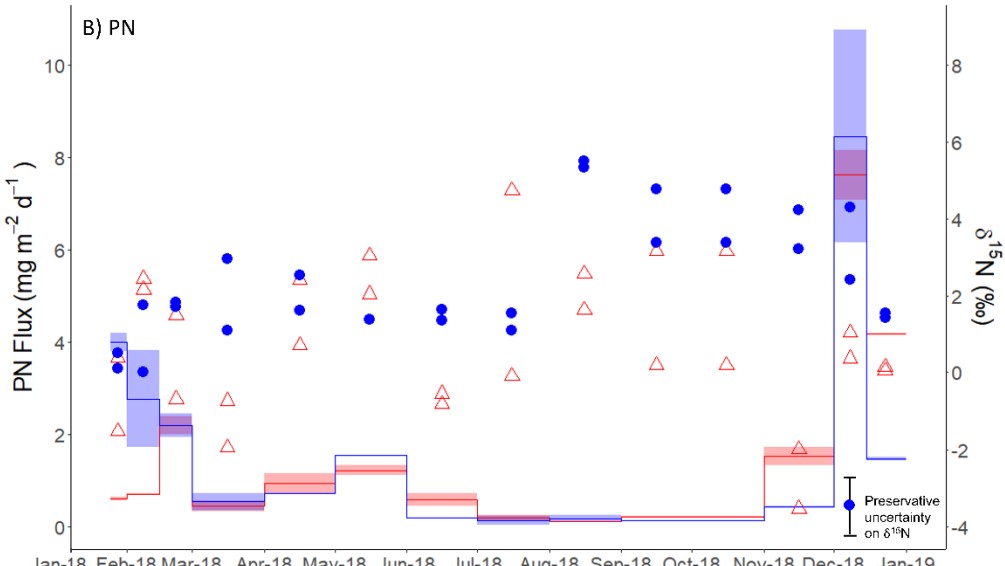

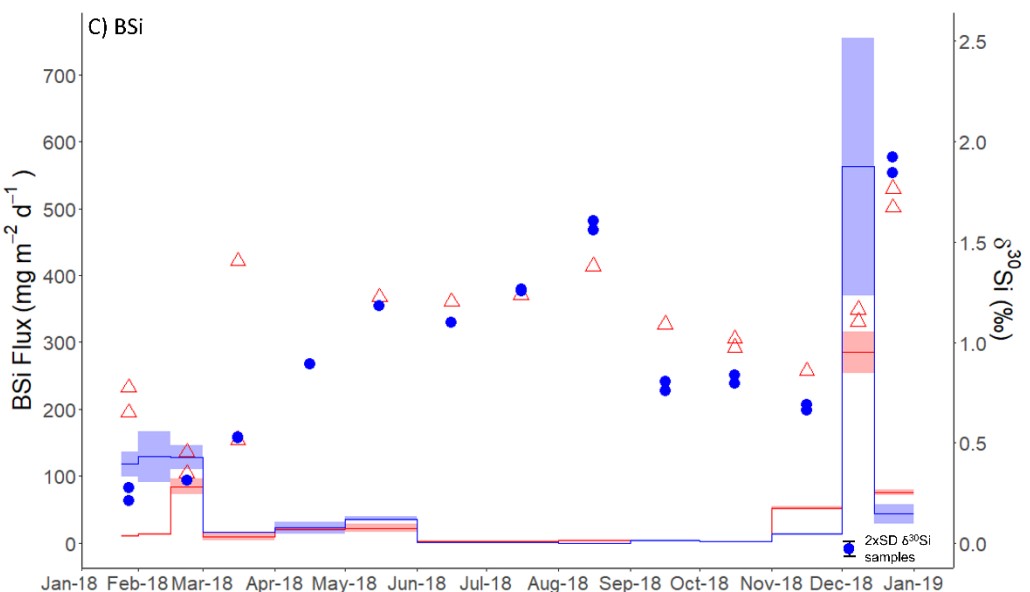


 *Figure 3: A) Particulate organic carbon (POC), B) particulate nitrogen (PN) and C) biogenic silica*
*(SiO$_2$, BSi) fluxes (mg m$^{-2}$ d$^{-1}$) at deep (red shading) and shallow (blue shading) sediment traps.*
*Shading indicates the maximum and minimum flux from two splits, with the solid line indicating*
*the mean value. Coloured points show isotope ratios for A) δ$^{13}$C$_{POC}$, B) δ$^{15}$N$_{PN}$ and C) δ$^{30}$Si$_{BSi}$ with red*
*open triangles and blue filled circles indicating deep and shallow sediment traps, respectively. The*
*legend shown in the top left hand corner of panel A applies to all panels. The maximum error on*
*sediment trap δ$^{13}$C$_{POC}$ (±1 ‰) and δ$^{15}$N$_{PN}$ (±1.5 ‰) values are shown by scaled error bars in the*
*bottom right corner, and are associated with formaldehyde preservation (Mincks et al., 2008) since*
*this vastly exceeds analytical error. For δ$^{30}$Si$_{BSi}$, the scaled error bar represents 2 x SD (0.07 ‰) for*
*the analytical sample replicates. For each sample, isotope ratios are given at the midpoint of the*
*period that the sample cup was open.*

3.3.  δ$^{13}$C$_{POC}$, δ$^{15}$N$_{PN}$ and δ$^{30}$Si$_{BSi}$ Isotopes

δ$^{13}$C$_{POC}$ values of deep and shallow sediment trap samples track each other well and show the same order of enrichment and depletion (Figure 3A). When describing the results for an individual sediment trap bottle, we give the mean of replicate splits from that sediment trap bottle unless otherwise stated. Initially, from January to March 2018, we see isotopically light δ$^{13}$C$_{POC}$ values between -27.40 and -28.56 ‰, before increasing to -24.38 ‰ and -25.07 ‰ in June in shallow and deep traps respectively. Over winter, δ$^{13}$C$_{POC}$ became more depleted (shallow: -25.76 ‰ in October, deep -27.07 ‰ in August) with a slight divergence (2.17 ‰) in the tracking of deep and shallow δ$^{13}$C$_{POC}$ in August 2018. Coinciding with increasing chlorophyll concentrations, δ$^{13}$C$_{POC}$ became more enriched during the period September to December 2018 (-25.72 to -21.13 ‰ and -26.04 to -21.98 ‰ for shallow and deep traps respectively).

Comparison of flux-weighted δ$^{13}$C$_{POC}$ values confirms the carbon isotopic similarity of deep and shallow traps, particularly during period 2 (Table 1). These results also highlight the shift in both δ$^{13}$C$_{POC}$ and δ$^{30}$Si$_{BSi}$ between period 1 and period 2.

*Table 1: Sediment trap seasonal (Jan 2018 – Dec 2018), period 1 (Jan 2018 – start of April 2018),*
*period 2 (Sept 2018 – end of Dec 2018), and winter (April – end of August) flux-weighted mean*
*δ$^{13}$C$_{POC}$ (‰), δ$^{15}$N$_{PN}$ (‰) and δ$^{30}$Si$_{BSi}$ (‰) for shallow (400 m) and deep (2000 m) traps. Given that*
*the analytical conditions were the same for all samples measured, we use the pooled variance over*
*the applicable time period as a measure of uncertainty on these mean isotopic ratios. Degrees of*
*freedom (dof) are based on cups with replicate isotopic measurements and are given in*
*parentheses.*

| Time period | δ$^{13}$C$_{POC}$ (‰) | | δ$^{15}$N$_{PN}$ (‰) | | δ$^{30}$Si$_{BSi}$ (‰) | |
|---|---|---|---|---|---|---|
| | Shallow | Deep | Shallow | Deep | Shallow | Deep |
| Seasonal | -25.15 ±0.49 (dof=14) | -24.40 ±0.45 (dof=14) | 2.07 ±0.34 (dof=14) | 0.39 ±0.43 (dof=14) | 0.50 ±0.09 (dof=8) | 0.86 ±0.10 (dof=6) |
| Period 1 | -28.59 ±0.34 (dof=4) | -27.24 ±0.41 (dof=4) | 0.98 ±0.40 (dof=4) | 0.15 ±0.66 (dof=4) | 0.21 ±0.109 (dof=2) | 0.59 ±0.16 (dof=2) |
| Period 2 | -22.47 ±1.03 (dof=5) | -22.79 ±0.74 (dof=5) | 2.97 ±0.66 (dof=5) | -0.09 ±0.65 (dof=5) | 1.54 ±0.30 (dof=4) | 1.08 ±0.14 (dof=4) |

| Winter | -25.31 ±0.63 (dof=5) | -26.25 ±0.39 (dof=5) | 1.81 ±0.49 (dof=5) | 1.74 ±0.64 (dof=5) | 0.58 ±0.20 (dof=2) | 0.48 ±0.17 (*) |
|--------|----------------------|----------------------|---------------------|---------------------|---------------------|----------------|

* There were no replicates from the deep sediment trap sample bottles for Si isotopes during this
period.
$\delta^{15}N_{PN}$ values are less consistent between deep and shallow sediment trap samples and there is
more heterogeneity between sample splits. For the shallow trap we see values ranging between
+0.13 and +2.96 ‰ (mean +1.42 ‰, SD 0.79 ‰) from January to June 2018, and, for the deep trap,
values ranged between -1.95 and +3.04 ‰ (mean +0.60 ‰, SD 1.60 ‰) during this period. Values
increase between June and August, reaching +5.42 and +2.10 ‰ in shallow and deep traps
respectively. From August to December (shallow), and August to November (deep), we see a trend
of decreasing $\delta^{15}N_{PN}$ to +1.49 and -2.77 ‰ in shallow and deep traps respectively, with the decrease
being of similar magnitude (3.93 and 4.87 ‰ respectively) for both traps. Shallow $\delta^{15}N_{PN}$ is
consistently higher than deep $\delta^{15}N_{PN}$ by 4.52 ‰ on average during this period (August to November).
In the deep trap we see a final increase in $\delta^{15}N_{PN}$ coinciding with the increase in PN flux from
November to December 2018, reaching a mean of +0.71 ‰. The same increase in $\delta^{15}N_{PN}$ is not
apparent in the shallow trap.
Si isotope compositions in deep and shallow samples were quite similar, exhibiting the same
seasonal patterns. Both deep and shallow traps showed an increase in $\delta^{30}Si_{BSi}$ from January to July
2018 (+0.71 to +1.24 ‰ in the deep trap, and +0.24 to +1.26 ‰ in the shallow trap) with the
steepest increase occurring from March to May (Figure 3C). Sample splits generally showed good
agreement with one exception during March 2018 when sample splits from the deep sediment trap
were +0.52 and +1.41 ‰, highlighting the heterogeneous nature of the sediment trap material.
Isotopic values in the deep trap were then quite steady over winter compared to the rest of the
record, with an increase of 0.38 ‰ in the shallow trap between May and August. At the end of
August, $\delta^{30}Si_{BSi}$ began to decrease steeply, reaching +0.68 and +0.86 ‰ in shallow and deep traps
respectively in November 2018. Following this, $\delta^{30}Si_{BSi}$ increased rapidly to +1.72 (deep) and +1.89 ‰
(shallow) coinciding with the large increase in BSi fluxes at this time.
### 3.4. Phytoplankton community structure
Eight samples (four deep and four shallow, Table 2) were analysed by light microscopy for
phytoplankton composition to cover the high productivity periods 1 and 2. Diatoms, silicoflagellates
and dinoflagellates were observed, with a dominance of diatoms (>85% by both abundance and
biovolume). Micro-zooplankton were also recorded, in particular radiolaria and tintinnids, though
these were not dominant by biovolume or abundance. Only intact cells were identified and counted.
In terms of abundance, during period 1, the diatoms *Fragilariopsis spp.* dominated both deep (58-66
%) and shallow (~70 %) trap samples (Figure 4A, C), whereas during period 2 the phytoplankton
community structure was more mixed with contributions from the diatoms *Thalassionema*
*nitzschioides, Chaetoceros,* small (<20 µm) centrics, as well as *Fragilariopsis spp.* Large centric
diatoms (>20 µm) represented 15-20 % of the community by abundance in the deep trap during
productive period 1, but <2.5 % in productive period 2. Interestingly we do not see these large
centrics in the shallow trap during productive period 1, implying that sinking velocities were < 76 m
$d^{-1}$ for these large phytoplankton cells based on the duration that the first sediment trap bottle was
open and the depth between the two traps.

In terms of biovolume, *Fragilariopsis spp.* were still a dominant component of the shallow trap sample in period 1 (~33 %) but were <9 % of the community in the deep trap during period 1, with the large cells of the diatom *Coscinodiscus* dominating with 39-67 % (Figure 4B, D). Diatoms, *Corethron pennatum* (shallow: 10-13 %; deep: 15 %)*, Rhizosolenia* (shallow: 9-21 %), and large centric diatoms (>20 µm) (shallow: 10-17 %; deep: 16-20 %)*, as well as the silicoflagellate *Dictyocha* (shallow: 9-10 %; deep: 8 %), were also relatively high in terms of biovolume during period 1. During period 2, the community was quite mixed in terms of biovolume in the shallow trap (Figure 4B). The deep trap had similar contributions from *Fragilariopsis spp.* (22-28 %), *Dictyocha* (14-15 %)*, Coscinodiscus* (10 %), and small (<20 µm, 9-14 %) and large (>20 µm, 9-19 %) centric diatoms during period 2. Since there has been little known work on the $\delta^{30}$Si of *Dictyocha* or indeed other silicoflagellates, we are not able to constrain the impact of this organism on our measured values. However, since the contribution by abundance was <5 % and diatoms were dominant (>85 %), their isotopic signature would need to be vastly different from that of diatoms to have an appreciable impact on our results.

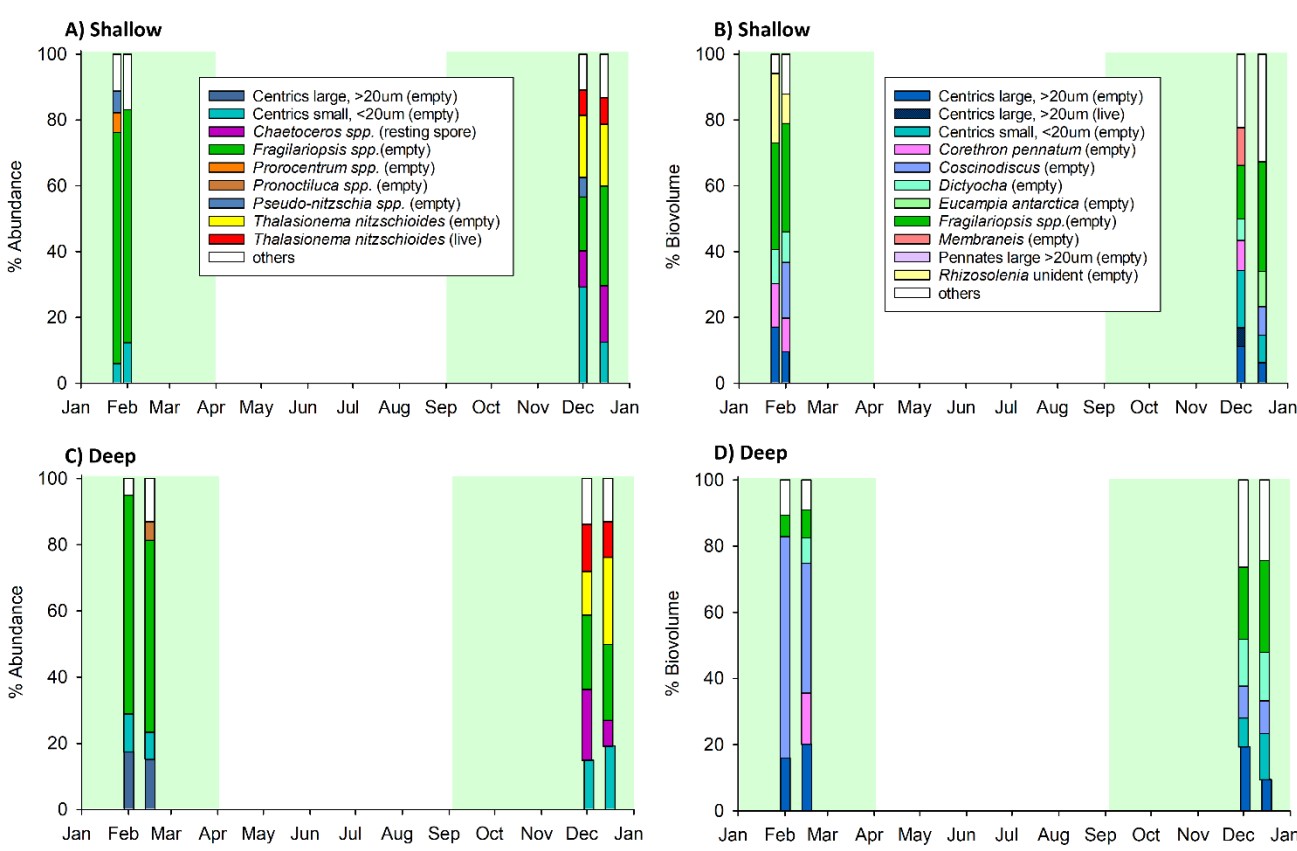

*Figure 4: Phytoplankton assemblage of A,B) shallow and C, D) deep sediment trap samples, according to abundance (A, C) and biovolume (B, D). Plots A and C show phytoplankton contributing >5 % by abundance, and plots B and D show >5 % by biovolume. Other refers to all other counted taxa combined. Four samples were identified taxonomically for each trap. Green shading highlights productive period 1 and 2, as per figure 2. Note that only intact cells were counted.*

## 4. Discussion

In this study we measure the seasonal cycle of POC, PN and BSi fluxes as well as the $\delta^{13}C_{POC}$, $\delta^{15}N_{PN}$ and $\delta^{30}Si_{BSi}$ values of sinking particles collected in shallow (400 m) and deep (2000 m) sediment traps in the Scotia Sea, Southern Ocean. Both the magnitude of fluxes and isotopic compositions were generally similar in the shallow and deep sediment traps, suggesting that most remineralisation occurred in the upper 400 m. This highlights that material reaching 400 m likely facilitates the transfer of carbon much deeper in the ocean, sequestering carbon for longer time periods (Kwon et al., 2009).

### 4.1. Seasonal flux cycles

The seasonal cycles of POC agree well with previously published work at the same location (Manno et al., 2015), with peaks in austral spring and late summer, though the peak POC fluxes recorded here (45.7 mg C m$^{-2}$ d$^{-1}$ and 43.4 mg C m$^{-2}$ d$^{-1}$, in shallow and deep traps respectively) are higher than those observed in previous years (22.9 mg C m$^{-2}$ d$^{-1}$; Manno et al., 2015). A smaller additional peak in POC flux (<10 mg C m$^{-2}$ d$^{-1}$) occurred in April/May, in agreement with some previous years (Manno et al., 2015). PN fluxes followed the same seasonal trend as POC for both deep and shallow traps suggesting a similar source. The similar magnitude of POC:PN ratios in period 1 in the two traps support consistency in the degree of degradation at these depths. The lower POC:PN ratios measured in the deep trap between August and October, compared to the shallow trap are consistent with a divergence in $\delta^{15}N_{PN}$ ratios, and could indicate that material arriving at the two traps is not necessarily sourced from the same region and time period in surface waters. Given the slower sinking speeds at this low-productivity time of year, it is possible that material reaching the deep trap is sourced from upstream of where material reaching the shallow trap is sourced in the regional circulation system. Different source regions are likely characterised by different phytoplankton assemblages with different nutrient stoichiometry, and the time taken for source material to reach each of the traps may well lead to differences in degradation state of organic matter, which could also lead to variations in POC:PN.

Our measured fluxes of BSi are higher than previously observed at this site at 2000 m (Rembauville et al., 2016). Maximum fluxes of 46.0 mg SiO$_2$ m$^{-2}$ d$^{-1}$ were recorded by Rembauville et al. (2016) in January 2012, which though of similar magnitude to our summer peak of 84.3 mg SiO$_2$ m$^{-2}$ d$^{-1}$, is an order of magnitude lower than the spring peak of 285.4 mg SiO$_2$ m$^{-2}$ d$^{-1}$ in December 2018. However, the Rembauville et al. (2016) record ends in November and therefore would not have captured the main peak in particle flux following the phytoplankton spring bloom in December (apparent in satellite surface chlorophyll; Figure 2 in Rembauville et al.(2016)). Additionally, we do not capture the first 3 weeks of January in our data. Interannual variability in export flux can be high due to the complexity of processes controlling the magnitude of export flux, such as community structure, nutrient limitation and zooplankton activity. Closset et al. (2015) measured very high fluxes (>700 mg SiO$_2$ m$^{-2}$ d$^{-1}$) of BSi south of the Sub-Antarctic Front in the Australian sector of the Southern Ocean at 2000 m, and similarly high fluxes have been observed in other sectors (Fischer et al., 2002; Honjo et al., 2000). A study by Trull et al. (2001) measured fluxes of BSi in the range of 30- 160 mg SiO$_2$ m$^{-2}$ d$^{-1}$ during the productive season in the same region as Closset et al. (2015), again highlighting the high interannual variability.

We define two main productive periods; productive period 1 from January to the start of April 2018, and productive period 2 from September to the end of December 2018, when chlorophyll concentrations were >0.4 mg m$^{-3}$. Satellite data suggest the magnitude of chlorophyll concentration was similar during both productive periods, but increasing in magnitude throughout period 2, and decreasing in period 1, consistent with timing of sampling. The particle fluxes associated with productive period 2 were much higher than those during productive period 1; a difference that is particularly pronounced for BSi fluxes. The bloom during period 2 was more geographically widespread (Figure S2) and thus it is possible that if more of the material reaching the trap was sourced from productive waters, this could have supported the higher fluxes observed at this time. The observed higher BSi fluxes in productive period 2 could also relate to the presence of more heavily silicified diatom species at this time, including the occurrence of resting spores (*Chaetoceros* spp*.;* Figure 4, and Rembauville et al. (2016)), increased aggregation (and thus sinking) potential, higher sinking rates, and/or reduced grazing pressure. The fact that we observed resting spores at the end of productive period 2 suggests that nutrients may have started to become limiting for at least some of the phytoplankton community (e.g. silicic acid and/or iron; Rembauville et al., 2016). POC and BSi fluxes track each other closely and ratios suggest substantial export of biogenic silica (Figure 5). This, combined with our visual observations of a dominance of phytoplankton material in the trap during the spring peak that was dominated by diatoms (Figure 4), suggest an important role for diatoms in transferring organic carbon to the deep ocean at this time. This could be achieved if cells are large, through large mineral (silica) ballasted cells sinking at high velocities (Baumann et al., 2022), or through the bioprotection of internal organic matter from grazing and oxidation by the diatom silica frustules (Passow and De La Rocha, 2006; Armstrong et al., 2001; Smetacek et al., 2004).

### 4.2. Seasonal variations in isotope ratios

Despite the strong relationship between particulate fluxes of POC and BSi, the relationship between the $\delta^{13}C_{POC}$ and $\delta^{30}Si_{BSi}$ isotope signatures is less pronounced (linear regression: $R^2 = 0.427$, p<0.001; Figure 5). This may relate to greater variation in the fractionation factor for $\delta^{13}C$ compared to $\delta^{30}Si$ (Brandenburg et al., 2022), as well as differences in remineralisation of organic carbon and silicon in the frustule. Additionally, whereas most of the $\delta^{30}Si$ signal is from diatoms, the $\delta^{13}C$ signal in the sediment trap material is also impacted by the presence of other organic material, e.g. zooplankton faecal pellets. We do not find significant relationships between $\delta^{15}N_{PN}$ and $\delta^{13}C_{POC}$ (p = 0.63) or $\delta^{30}Si_{BSi}$ (p = 0.60). We discuss results for each of the 3 main periods, productive period 1 (first export event), the winter flux hiatus, and productive period 2 (second export event).

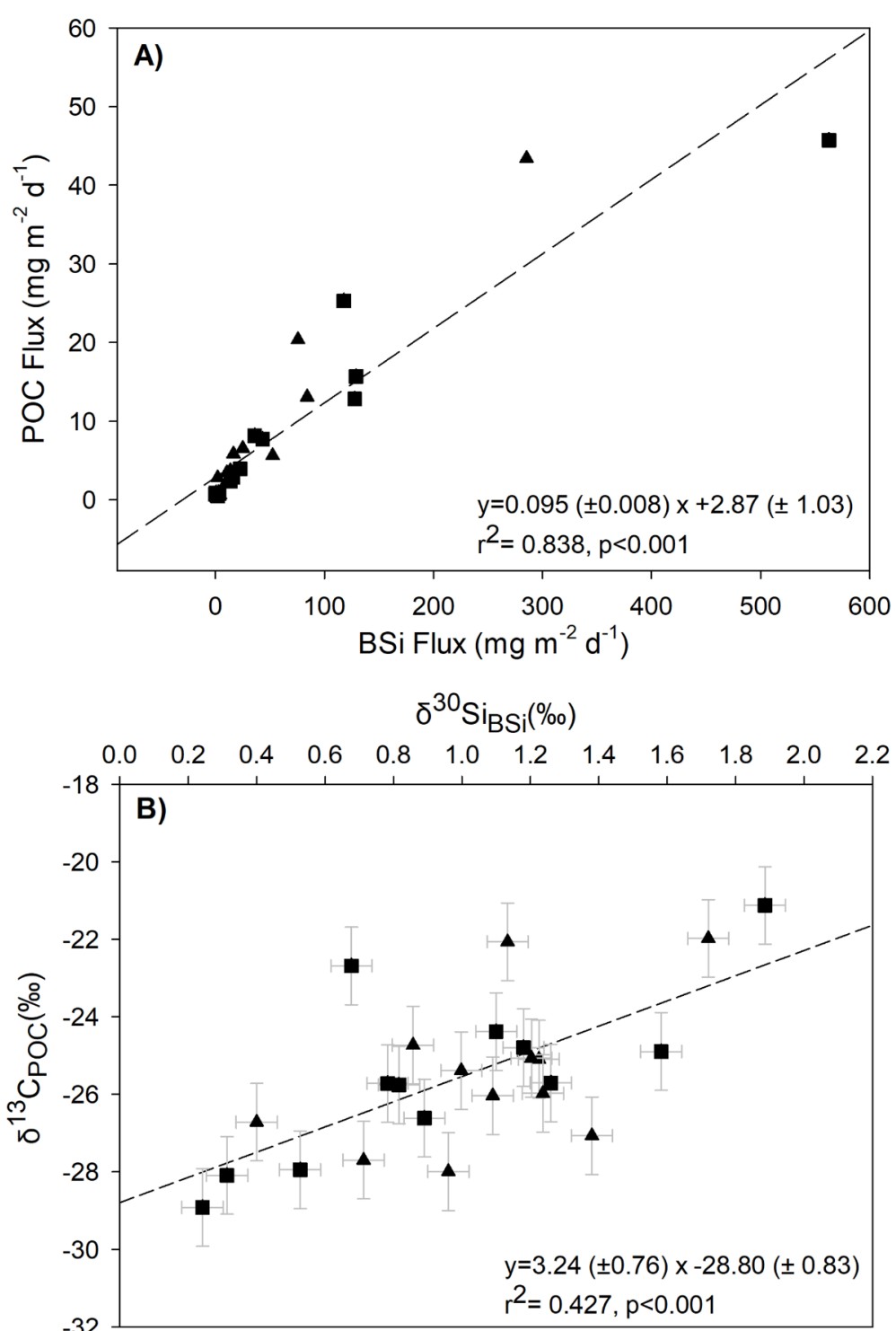

517

*Figure 5: Relationship between BSi and POC for data from both deep (triangles) and shallow*
*(squares) sediment traps. A) Regression between BSi and POC fluxes, and B) between $\delta^{13}C_{POC}$ and*
*$\delta^{30}Si_{BSi}$. Regression lines are shown by dotted lines with coefficients and associated standard errors*
*also shown. Error bars on isotope values represent the maximum error on sediment trap $\delta^{13}C_{POC}$*
*(±1 ‰) associated with formaldehyde preservation (Mincks et al., 2008) and for $\delta^{30}Si_{BSi}$, the scaled*
*error bar represents 2 x SD (0.07 ‰) for the analytical sample replicates.*

*4.2.1. Productive period 1*

During productive period 1, $\delta^{13}C_{POC}$ is low, averaging -28.59 and -27.24 ‰ in shallow and deep traps
respectively, close to that expected for Southern Ocean phytoplankton employing typical C3
metabolism (i.e. diffusive $CO_2$ transfer into the internal cell pool and Rubisco carboxylation) (Raven,
1997). This is consistent with the dominance of diatoms (*Fragilariopsis spp.*) in the trap material, as
Bacillariophyceae are known to employ C3 metabolism (Table IV in Raven, 1997)**.** Preferential uptake
of $^{28}Si$ by diatoms (De La Rocha et al., 1997) during the late spring bloom of productive period 1 also
explains the low $\delta^{30}Si_{BSi}$ values. BSi:POC ratios were elevated at the start of productive period 1,
which may suggest that phytoplankton were heavily silicified. The contribution of non-siliceous
phytoplankton was low during the periods analysed for phytoplankton composition (<2%, with the
exception of the shallow trap in the late January sample where the contribution was 6.7%), though
we cannot rule out higher contributions of non-siliceous phytoplankton during other periods which
could account for the lower BSi:POC ratios at these times. After initial low values, we see a
progressive increase in both $\delta^{13}C_{POC}$ and $\delta^{30}Si_{BSi}$, reflecting the progressive utilisation of both $^{12}C$ and
$^{28}Si$ as nutrient pools are consumed during the bloom. As such, the diatom cells reaching the
sediment trap in late spring/summer were utilising increasingly isotopically-enriched C and Si for
growth leading to progressive isotopic enrichment of the cells sinking into the sediment trap. This
observation fits with elevated but decreasing surface chlorophyll concentrations from February to
April 2018. Increasing $\delta^{13}C_{POC}$ and $\delta^{30}Si_{BSi}$ into the late summer may also partially reflect preferential
remineralisation of the more labile $^{12}C$ and $^{28}Si$ in particles as they sink through the upper 400 m of
the water column. The lack of variation in $\delta^{13}C_{POC}$ and $\delta^{30}Si_{BSi}$ between 400 and 2000 m in our study
suggests that remineralisation may be limited between these depth, or that there is no further
fractionation effect. Whilst laboratory-based silica dissolution experiments are equivocal (Demarest
et al., 2009; Wetzel et al., 2014), our findings agree with field studies that also indicate a lack of Si
isotopic fractionation during diatom silica dissolution (Closset et al., 2015; Egan et al., 2012).
During productive period 1 there was no clear trend in $\delta^{15}N_{PN}$, with values between -1.95 and +2.96
‰. We speculate that this mixed signal resulted from a combination of surface phytoplankton using
both ammonium and nitrate as the inorganic nitrogen source, and variability in the sediment trap
material composition. Enrichments of 2-4 ‰ occur between successive trophic levels, and egestion
and excretion can have varying isotopic effects (see Section 4.3), thus the presence of faecal pellets,
animal moults and carcasses could alter the isotopic composition of the sediment trap material.
Additionally, any supply of ammonium through remineralisation would be utilised quickly because
ammonium is kinetically favourable to nitrate (Glibert et al., 2016), resulting in particles with a
decreased $\delta^{15}N_{PN}$ compared to those produced by nitrate assimilation.

*4.2.2. Winter hiatus*
Between May and August, both $\delta^{13}C_{POC}$ and $\delta^{30}Si_{BSi}$ showed little change, with a slight progressive
decrease for $\delta^{13}C_{POC}$ and increase in $\delta^{30}Si_{BSi}$. It is possible that the slight progressive trend towards a
lighter carbon isotopic composition of sinking particles from -24.94 to -25.98 ‰ is driven by a
mixture of older, isotopically heavier particles that have undergone partial remineralisation and the
input of material of different isotopic composition from the small secondary peak in POC we
observed in April/May. An input of smaller, more slowly sinking cells reaching the trap in increasing
numbers following the initial late spring peak in production could drive the lower $\delta^{13}C_{POC}$ at this time.
Additionally, the pulse of material could be driven by a successive peak in production of a different
phytoplankton community with a different isotopic signature. Korb et al. (2012) found an increasing
presence of dinoflagellates from spring to summer, as well as seasonal changes in the size structure
of the phytoplankton community to the northwest of South Georgia, supporting either hypothesis.
We do not have the species composition data from this time period to evidence this directly, but we
suggest that the reduction in $\delta^{13}C_{POC}$ does not relate to a mixing event and a resupply of $^{12}C$, due to
the fact that $\delta^{30}Si_{BSi}$ continued to increase slowly. Given the generally lighter silicon isotopic
composition of seawater below the photic zone, we would expect a mixing event to also result in a
decline in seawater $\delta^{30}Si$ and consequently $\delta^{30}Si_{BSi}$. This would mean that our hypothesised shift in
phytoplankton species composition in the traps (May-August) did not impact Si fractionation to the
same extent as carbon isotopes. Whereas size, growth rates, cell geometry and different carbon
acquisition mechanisms have all been highlighted as impacting the $\delta^{13}C_{POC}$ of marine plankton (Popp
et al., 1999, 1998; Bidigare et al., 1999; Trull and Armand, 2001; Tuerena et al., 2019), species-
dependent Si fractionation by polar and subpolar diatoms has only been observed in the laboratory,
not in the field (Annett et al., 2017; Cassarino et al., 2017; Sutton et al., 2013). $\delta^{15}N_{PN}$ in the shallow
trap showed a slight progressive decrease from April to July, before increasing in August to 5.42 ‰.
The progressive decrease is consistent with the propagation of the surface signal of phytoplankton
growth and fractionation, with a longer time lag than during spring and summer due to slower
sinking rates during the low-productivity period. Decreasing $\delta^{15}N_{PN}$ reflects the increasing influence
of ammonium uptake, either in the same locale or upstream in the regional circulation system,
which leads to lower $\delta^{15}N_{PN}$ than nitrate uptake in the slowly-sinking flux. The large range in $\delta^{15}N_{PN}$ in
the deep trap in July makes it difficult to determine with certainty a trend in $\delta^{15}N_{PN}$ in the deep trap
between July and October. Dissimilar trends in $\delta^{15}N_{PN}$ between the two traps over the winter period
also support the argument that material reaching these two traps may have a different source
region or time period in surface waters (Section 4.1).

*4.2.3. Productive period 2*
At the start of productive period 2 (September) we saw a significant decrease in $\delta^{30}Si_{BSi}$ (~0.5 ‰) in
both traps suggesting resupply of $^{28}Si$ enriched silicic acid to the euphotic zone via mixing.
Interestingly, we did not see the same consistent shift in carbon isotopes; we measured a ~1 ‰
decrease in the shallow trap $\delta^{13}C_{POC}$ and a ~1 ‰ increase in the deep trap $\delta^{13}C_{POC}$. We speculate that
this mixing could bring waters of increased silicic acid concentrations to the surface, promoting full
expression of the isotope fractionation effect from phytoplankton uptake and thus lower $\delta^{30}Si_{BSi}$ in
sinking particles. To match our observations, these mixed waters would need to be similar in
dissolved inorganic carbon concentrations and $\delta^{13}C$, which could relate to the depth of mixing and
differences in the depth at which POC and BSi are remineralised (Friedrich and Rutgers van der Loeff,
2002; Weir et al., 2020). We note that current velocities recorded at this time were elevated (Figure
S1), particularly in the deep trap, suggesting a shift in the surrounding velocity fields, which may
have resulted in biased sample collection at this time through either over- or under-collection
(Buesseler et al., 2007). Whereas $\delta^{13}C_{POC}$ progressively increased during productive period 2, from -
25.88 ‰ in September to -21.56 ‰ at the end of December (mean of deep and shallow traps),
$\delta^{30}Si_{BSi}$ continued to decrease until November before showing a sudden increase from +0.74 ‰ to
+1.80 ‰ at the end of the sampling period. This may suggest that DSi, or co-limiting nutrients, was
replete, and uptake could occur unhindered until November 2018 when very high rates of
production and the associated high fluxes of BSi increased the demand for DSi and led to enrichment
of $\delta^{30}Si$ in overlying waters and subsequently sinking siliceous phytoplankton. For carbon, uptake
was sufficient from September to progressively deplete source waters in $^{12}C$, driving an increase in
$\delta^{13}C$ in surface waters and newly formed phytoplankton cells. BSi:POC ratios increased from
September to December suggesting that material reaching the traps was increasingly silicified.
Interestingly, unlike C and Si isotopes, we saw a divergence in the nitrogen isotopic composition of
deep and shallow traps between August and December. The sharp increase in mean $\delta^{15}N_{PN}$ from
+1.32 ‰ in July to +5.42 ‰ in August 2018 in the shallow trap that initiated the divergence strongly
suggests an advective change in source material. As noted above, this was a period of increased
horizontal velocities and may have facilitated material reaching the two traps from different sources
of differing initial composition and degradation states. The substantially lower $\delta^{15}N_{PN}$ in the deep
trap from August to November, compared to that of the shallow trap is surprising. It would be
expected, that, as particles sink and are progressively decomposed this would remove dissolved
nitrogen depleted in $^{15}N$, thus increasing $\delta^{15}N_{PN}$ in the particles. Indeed many studies have observed
this trend of increasing $\delta^{15}N$ with depth in suspended particles (Altabet et al., 1991 and references
therein). However, like Altabet et al. (1991), we observe lower $\delta^{15}N_{PN}$ in sinking particles in the deep
sediment trap. This has also been observed previously in Antarctic waters (Wada et al., 1987).
Though the reason for this is not well understood (Sigman and Fripiat, 2019), it appears to be a
consistent phenomenon. Particles in our deep trap must therefore be gaining light nitrogen or losing
heavy nitrogen and could reflect a different source composition. In agreement with Altabet et al.
(1991), we suggest that lateral transport of low $\delta^{15}N_{PN}$ from a region of increased ammonium-based
production could explain this, highlighting a difference in the source of sinking particles to the two
traps. Altabet et al. (1991) also suggests that, since protein nitrogen is 3 ‰ higher than bulk
nitrogen, the selective decomposition of protein could explain the decrease in $\delta^{15}N$ with depth,
though why this would not be the case also for suspended PN is unclear. We observe the greatest
divergence in shallow and deep N isotope compositions during periods of low PN flux (Figure 3),
consistent with the observations of Altabet et al. (1991), enabling a low flux of laterally supplied
material to have an amplified impact on the isotope signal. In support of this, in December when
particle fluxes increase sharply with the spring bloom, $\delta^{15}N_{PN}$ in the deep trap increases more in line
with that of the shallow trap, highlighting a switch from source material being dominated by lateral
supply when vertical supply is negligible, to the dominance of vertical supply from surface
production following the phytoplankton bloom.

*4.3. Drivers of shifting isotopic ratios*
The mean flux-weighted isotopic composition measured during productive periods 1 (January to the
start of April 2018) and 2 (September to the end of December 2018) suggests that the processes
driving the flux of material at these times differ (Figure 3, Table 1). The divergence in the $\delta^{15}N_{PN}$ of
deep and shallow trap material during period 2 limits our ability to compare the temporal shifts in
mean isotopic ratios for nitrogen isotopes, so we focus here on $\delta^{13}C_{POC}$ and $\delta^{30}Si_{BSi}$. Since our record
does not extend beyond December 2018, and we do not capture the first 3 weeks of January 2018
when fluxes were likely high, we do not record the initial value at this time, however, we would
expect $\delta^{13}C_{POC}$ to be even more negative at this time. We cannot determine if $\delta^{13}C_{POC}$ and $\delta^{30}Si_{BSi}$
would return to values akin to that in period 1 in the following late spring-summer season (January
2019). We saw a shift in $\delta^{13}C_{POC}$ from a mean of -28.31 ‰ in January 2018 at the time of our first
measurements to -25.88 ‰ in September at the start of period 2. This coincided with a change in
community structure, with abundance dominated by *Fragilariopsis spp.* in period 1 to a more mixed
community in period 2. Of the abundant phytoplankton species (>5%, Figure 4A, C), we find
statistically significant linear relationships between $\delta^{13}C_{POC}$ and percent abundance for *Fragilariopsis*
*spp.* (empty: $R^2 = 0.926$, p<0.001), *Thalassionema nitzschioides* (live: $R^2 = 0.774$, p=0.004; empty: $R^2 =$
0.844, p=0.001), and *Chaetoceros spp. (resting spore)* ($R^2 = 0.732$, p=0.007). We stress this is based
on only 8 samples. Nevertheless, these robust samples show that there was a shift in phytoplankton
community structure. Though *Fragilariopsis spp.* were mainly empty cells, colonisation by bacteria
(Grossart et al., 2003; Kiørboe et al., 2003) may facilitate carbon transfer within and on these cells,
and certainly the live cells of *T. nitzschioides* and resting spores of *Chaetoceros spp.* would act as
agents of carbon transfer (Agusti et al., 2015; Salter et al., 2012; Rembauville et al., 2016).
We examine whether this shift in phytoplankton community composition is associated with a change
in SA:V (Table 2) since greater fractionation of carbon in smaller phytoplankton cells with higher
SA:V is well observed in the literature (e.g. Popp et al., 1998; Tuerena et al., 2019). There was a
statistically significant (paired t-test, p=0.008) difference in the community SA:V between productive
periods, increasing from 0.35 $\mu m^2$ $\mu m^{-3}$ in period 1 to 0.51 $\mu m^2$ $\mu m^{-3}$ in period 2. However, this would
result in increased isotopic fractionation during period 2, which is the opposite to what we observed.
We note here, that as only intact cells were counted, the measured SA:V ratios may not fully account
for the isotopic composition of the trap material due to the presence of fragmented material. It is
possible that there was a change in the mechanism of carbon uptake with the more mixed
phytoplankton community in period 2 using $HCO_3^-$ instead of $CO_2$ or employing carbon concentrating
mechanisms (CCMs), both of which would result in higher $\delta^{13}C_{POC}$ than the diffusive uptake of $CO_2$
and Rubisco carboxylation (Raven, 1997; Cassar et al., 2004). Studies show that there is much
diversity amongst diatoms in the use of CCMs and many are able to take up both $CO_2$ and $HCO_3^-$
(Trimborn et al., 2009; Roberts et al., 2007; Shen et al., 2017; Young et al., 2016). We suggest that
species-driven differences in carbon uptake mechanisms account in part for the differing $\delta^{13}C_{POC}$ that
we observed during the two main productive periods.
***Table 2: Phytoplankton cell community surface area to volume (SA:V) ratios measured in deep and***
***shallow sediment traps for samples enumerated in both productive periods 1 and 2.***

| Bottle open date | Depth | Period | Mean community SA:V |
|---|---|---|---|
| 25/01/2018 | Shallow | 1 | 0.39 |
| 01/02/2018 | Shallow | 1 | 0.35 |
| 01/02/2018 | Deep | 1 | 0.33 |
| 15/02/2018 | Deep | 1 | 0.32 |
| 01/12/2018 | Deep | 2 | 0.53 |
| 01/12/2018 | Shallow | 2 | 0.48 |
| 15/12/2018 | Deep | 2 | 0.53 |
| 15/12/2018 | Shallow | 2 | 0.52 |

We also observed a shift in the mean flux-weighted $\delta^{30}Si_{BSi}$ ratios (Table 1) between period 1 and
period 2. With the exception of one culture study (Sutton et al., 2013), systematic species-driven
shifts in $\delta^{30}Si_{BSi}$ fractionation have not been observed (e.g., De La Rocha et al., 1997), suggesting that
there may be an additional driver of the changing isotopic ratios. Since, prior to our first
measurements there had been a long-lasting phytoplankton bloom (Figure S2), we would expect
production to have utilised much of the light $^{28}Si$, resulting in particles with enriched $\delta^{30}Si_{BSi}$ reaching
the trap in January 2018. However, we observe isotopically light mean values of +0.48 ‰ at the start
of sampling at the end of January, suggesting that there must have been a resupply of $^{28}Si$. Physical
mixing, bringing deep and benthic waters rich in nutrients, including iron, to the surface waters
around South Georgia, are known to support the large blooms occurring downstream of South
Georgia (Matano et al., 2020; Nielsdóttir et al., 2012) and could supply both $^{12}C$-enriched dissolved
inorganic carbon and $^{28}Si$-enriched silicic acid. Additional nutrients could also be supplied to our
study region by glacial discharge associated with isotopically light silicon isotopic signatures (Matano
et al., 2020; Hatton et al., 2019), or benthic fluxes from shelf sediments, likely also releasing
isotopically light DSi (Ng et al., 2020; Cassarino et al., 2020; Closset et al., 2022). Therefore, we
suggest that low values (increased fractionation) of $\delta^{13}C_{POC}$ and $\delta^{30}Si_{BSi}$ during period 1 relate to
increased nutrient availability enabling full expression of the isotopic fractionation and thus
isotopically light particulate material to reach the sediment trap.
The ocean circulation in our study region is complex and variable on fine spatial and temporal scales,
affecting horizontal and vertical velocities (e.g. Boehme et al., 2008). It is clear from the currents
measured at the depths of our two traps (Figure S1), that both the direction and magnitude of the
flow can vary within and between seasons and is not necessarily consistent between the two depths.
There are thus potentially different source regions for material in the two traps at certain times of
the year as suggested for example by $\delta^{15}N_{PN}$.ratios in winter. We lack the full depth resolution of
vertical and horizontal velocity fields and information on sinking rates to confirm this, but previous
studies have highlighted variability in the locations of the Southern Antarctic Circumpolar Current
Front and the Polar Front, as well as eddies generated from these fronts, in our study region (Moore
et al., 1999; Boehme et al., 2008; Whitehouse et al., 1996). We suggest that variability in ocean
current velocities could explain different isotopic ratios in period 1 and 2, through the supply of
material to the traps from different source regions with differing nutrient and remineralisation
regimes. Different source waters would impact nutrient availability including iron supply, uptake and
recycling (Hawco et al., 2021; Ellwood et al., 2020), which in turn influences species composition,
nutrient utilisation and uptake rates (e.g. Meyerink et al., 2019). This highlights the importance of
making synchronous, and full depth resolution measurements of physical processes such as current
strength and direction, to be able to distinguish between spatial and temporal drivers of shifts in
species composition, particle flux and isotopic composition.
Since trophic transfer is known to impact both carbon and nitrogen isotope compositions of organic
matter, the presence of moults and faecal pellets in trap samples is also important to consider. An
incubation study focussed on *Euphausia superba* found that the $\delta^{15}N$ of the *E. superba* faecal pellets
was always lower than that of the copepods they ingested, though still higher than that of POM
(Schmidt et al., 2003). Additionally, Tamelander et al. (2006) measured faecal pellets produced by
copepods with depleted $^{15}N$ compared to the algal food source. Though a few studies on temperate
and subtropical copepods showed that the faecal material had similar or slightly higher $\delta^{15}N$ than
the food source (Altabet and Small, 1990; Checkley and Entzeroth, 1985), there is not a consistent
fractionation effect of egestion for either $\delta^{15}N$ or $\delta^{13}C$, which may relate to compositional
differences (protein, carbohydrate, lipid) and their isotopic values (Tamelander et al., 2006). We are
therefore not able to determine the impact of faecal pellets or moults on the isotopic composition of
our samples. As phytoplankton material dominated at the times of peak flux, we suggest that the
importance of faecal pellets and moults may be greater during periods of lower flux, however we
cannot rule out their contribution during the bloom periods. We suggest that it would be highly
informative to conduct particle specific isotope analysis of common particle types in sediment traps,
such as faecal pellets, phytoplankton detritus and zooplankton moults, to improve our ability to
determine the impact of particle flux composition on bulk isotope compositions.

**Conclusion**
The seasonal cycles in primary productivity and nutrient uptake in surface waters at our study site in
the Scotia Sea are reflected in the fluxes and isotopic ratios of sinking particulate material. We find
that most remineralisation occurs in the upper 400 m of the water column and below this the
magnitude of the flux of sinking material is relatively consistent, supported by consistency in
POC:PON ratios. We find that particulate fluxes of C and BSi are tightly coupled which highlights the
importance of siliceous material in the transfer of POC to depth. We suggest that a change in
phytoplankton community structure can at least partly explain the shifts in carbon isotopic
composition between the two productive periods measured here. Though complex, seasonal
patterns in isotopic composition of particulate material reaching the sediment traps do reflect the
degree and type of nutrient utilisation in the source surface waters. Our data also suggest an
importance of laterally supplied material to the sediment traps and supports seasonal differences in
source regions. Our results highlight the need for more detailed mechanistic understanding of the
drivers of POC flux and biogeochemical cycling, to improve estimates of the current and future
strength of the biological carbon pump and the ocean's role as a $CO_2$ sink.

**Data availability**
Phytoplankton abundances and biovolume, as well as mean flux and isotopic ratios are available
with the following DOI's:
DOI in progress with the British Antarctic Survey Polar Data Centre
**Author contributions**
AB and CM conceived the study and participated in fieldwork to collect samples. AB conducted
laboratory analysis with support from TW, LF, and UD for isotope analysis. MW conducted
phytoplankton analysis and provided intellectual input on phytoplankton community composition.
SH and KH provided support for isotopic analysis and contributed to the interpretation of the data
and implications. CC supported uncertainty analysis. All authors contributed text to the manuscript.
**Competing Interests**
The authors declare that they have no conflict of interest.

**Acknowledgements**

We are very grateful to the scientists and crew aboard research cruises JR17002 and DY098 for their efforts to deploy and recover the P3 mooring. We thank staff at the Bristol Isotope Group for running and maintenance of the mass spectrometer facilities at the University of Bristol, as well as Colin Chilcott for technical support for C and N analysis at the University of Edinburgh. AB and CM were supported by NC-ALI funding and ecosystems programme. CM was also funded by UKRI FLF project MR/T020962/1. SH was supported by the United Kingdom Natural Environment Research Council through grant NE/K010034/1. UD was supported by the UK NERC through grant NE/P006108/1. LF was supported by a NERC GW4+ DTP studentship and TW by a CSC-UoB Joint Scholarship. We thank Sally Thorpe and Emma Young for insights on the physical oceanographic conditions of the region. Finally, a special thanks to Flo Atherden for her dedicated work picking out swimmers from the shallow sediment trap.

782

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
