# Peer review of "Seasonal cycles of biogeochemical fluxes in the Scotia Sea, Southern Ocean: A stable isotope"

_Biogeosciences, 2022_

## Referee Comment (RC1)

**General comments**

In "Seasonal cycles of biogeochemical fluxes in the Scotia Sea, Southern Ocean: A stable isotope approach", Belcher et al. present a study investigating the seasonal variations of organic matter (POC and PON) and biogenic silica fluxes from two sediment traps located north-West of South Georgia in the Scotia Sea (Southern Ocean). Using stable isotope approaches the authors examine the origin and some of the processes controlling the fluxes they have observed in the traps.

They investigated the differences between two productive events (in February 2018 – summer season – and December 2018 – spring season) and the coupling of C, N and Si fluxes during these events. Their main results are: Particulate fluxes and isotopic compositions were similar in the deep and shallow trap suggesting that most of the remineralization occurred in the upper layer of the water column. Despite a very noisy d15N signal, the synchronicity if the d30Si and d13C signals highlight the coupling between these two elements and the significant role of diatoms in the export of C (and BSi) in the area. Based on the estimation of isotopic baselines associated with the two productive events, they also suggested a change in the source region of the material coming into the sediment traps.

Generally speaking, the results of this study are interesting. However, I found the manuscript rather hard to read, often unclear or confusing. I think that the manuscript would profit from an effort to make the structure of the discussion more easily understandable for the reader. More work can also be done regarding the description of the analytical and sample processing methods as well as data quality. Some important information is missing and/or unclear. But most importantly, I think the authors should re-think figure 3, which is one of the most (if not the most) important figure of the manuscript. Indeed, the way the figure is built does not support or illustrate the statements or hypothesis authors are attempting to demonstrate. Additionally, there are also some inconsistencies in the wording, and I would strongly encourage the authors to carefully read their manuscript again and have it read by an English-speaking person.

I detail these points below, together with minor points that the authors should also consider while carefully revising this manuscript. I recommend publication of this paper in "Biogeosciences" after major revisions.

**Major concerns**
Currently, the manuscript requires very careful reading (and re-reading) in order to understand the authors' argumentation and get a sense of the various settings. A few suggestions:
* Re-organizing the discussion based on the three main periods that are discussed in the manuscript. For example, having three subsections in chronological order (i) Early spring event (P2), (ii) Late summer event (P1), (iii) Winter hiatus; or to fit with the main figure of the manuscript (fig 3) (i) First export event (P1), (ii) Winter hiatus, (iii) Second export event (P2).

* As it is, the manuscript needs desperately figures that will support the authors' hypothesis and statements for two main reasons: (i) Some important figures are missing. For example, figures illustrating the relationship between POC and BSi (mentioned for example L415) or d13C and d30Si (mentioned L422) with associated R2 and statistics. Right now, there is no figures to illustrate or support the relationship authors are discussing in the manuscript. (ii) Figure 3, the key figure of the manuscript, is currently very poorly designed. The choice of shading to represent fluxes in mg m-2 d-1 does not actually reflect the full magnitude of those fluxes. The most obvious example is POC flux in the shallow trap in May 2018. It seems to "peak" for a short period of time to values around 8 mg m-2 d-1 while it was sustaining this rate for a long period of time (31 days). Fluxes will appear less biased by using barplot representing the mean flux and error bar for the standard deviation. The variations of the isotopic composition of particles (d13C, d15N and D30Si) are also poorly illustrated by the choice of representing only the min and max on the figure. A mean value with error bar will be more representative of the seasonal evolution of the signal, as well as of the heterogeneity of the material (when error bars are more widely spread). It will also help with the scattering of the d15N and validate (or invalidate) the trends suggested by the authors.

* Something that need to be highlighted in the method section and briefly discussed later on is that the total collection period is 341 days. The first cup opened on Jan 25$^{th}$ (2018) and the last one closed on Jan 1$^{st}$ (2019). I would be worth it mentioning that the sediment trap series misses 3 weeks in the beginning of January where the flux is expected to be significant. Authors have not made any annual/seasonal integration of their fluxes, but they should still discuss the risk of missing a significant part of the seasonal flux early in the season. It might be of importance for the discussion regarding the isotope baseline for the first export event.

* It is not clear in the method section if authors have used the different splits of samples as replicates or if they have combined splits to do their different measurements. Figure 3 gives two values for each measurement (a min and a max) so one can guess that authors have measured duplicates out of those splits. Going through the discussion section, authors start to mention these mean values that do not correspond to anything presented in figure 3. I do not see the point of plotting only the min and max on the figure while using a mean value in the text. This is confusing, make things unclear and prevent the figure to illustrate and support correctly the text (e.g. it is hard to see some of the trends that are discussed in the text). I am suggesting using the mean values in the figure with the corresponding error bar and add those error (as standard deviation) within the text.

* Please define what "isotopic baselines" is. It is not defined anywhere in the manuscript, neither it is explained how authors have estimated the different baseline they are referring to. If it is just the lightest isotopic signature recorded just before a productive (and export) event, I am suspicious with the isotopic baseline identified for the first event since a significant part of the flux might have been missed early January 2018. Moreover, the isotopic baselines are not identified in figure 3 neither in Table 1. In general, this last part of the discussion (related to the comparison of the different isotopic baselines) is quite confusing and unclear and need some serious rephrasing.

**Minor concerns**

*Introduction*

* L43 Use biological pump of carbon instead of BCP.

* Sediment traps have bias too. A short summary (one-two sentences + ref) of them would be useful here. Especially since authors discuss some of them later in the manuscript.

* L73 Use "challenges" instead of "complications"

*Does sea-ice affect the region where the traps are located? It has been shown that the occurrence of sea ice can significantly affect stable isotopes composition (at least for d13C - e.g. Kennedy et al. 2002 - and d30Si - e.g. Fripiat et al. 2007). This could be an important aspect to consider in your discussion.

*The processes controlling the stable isotopic composition of C and N are quite well introduced but authors have been very quick concerning d30Si. More information about fractionation and the processes controlling it need to be introduced here (e.g. difference in fractionation between polar diatom species - Sutton et al. 2013 – fractionation (or not) associated with biogenic silica dissolution - Demarest et al. 2009, Wetzel et al. 2014).

*L115-116 what is the difference between annually and seasonally? Are they calculated over a different period?

* Do the stable isotopes really give information about the actual composition of organic matter?

* Perhaps what is missing in the introduction is a paragraph about why is the composition of particles important for the biological pump of carbon? For example, it will affect the sinking speed of particles (and authors discuss this later), their recycling in the water column (and later within the sediment) etc.

*Material and Methods*

* L152-153 it would be interesting to quantify this effect in percent of the signal

* L168 splitted

* It is not clear in the different paragraphs of the method section if these slips were combined or analyzed separately as replicates. If it is the second option (which is my guess) these replicates can be used to calculate the error or std on the samples as the potential heterogeneity of the sample will be reflected by a large error bar on the sample.

* L183 "per mil" instead of "per mille"

* Please define the meaning of "PACS international standard"

* L191 several statistical errors have been described here, although it is mentioned later in the manuscript, authors should indicate which one they have associated to their measurement).

* It would be interesting to quantify those splits in percent of the total sample

* What about lithogenic silica? Alkaline extraction method using NaOH will dissolve some lithogenic material along with BSi (Ragueneau et al. 2005). Because LSi has a light d30Si (down to -2.3 pmil, Opfergelt and Delmelle 2012), it has the potential to bias BSi d30Si measurements even with low LSi contribution (or contamination) to the alkaline digestion. For example, and in the worst-case scenario of LSi d30Si of -2.3 pmil, a contribution of 3% during period 2 and 4% during period 1 would significantly bias the result (by 0.1pmil). This is a rough calculation, but this need to be discussed as sediment traps can collect a significant amount of lithogenic

material. Nota that methods have been proposed to "correct" BSi d30Si from LSi contamination (see Closset et al 2015).
* L206 HCL or Milli-Q water as eluent?
* L222 This sentence is unclear. what are those pseudo replicates? two samples per pseudo replicates so four isotopic measurement per bottle? Isn't the case for all isotopic measurements (and for POC and PON fluxes too)?
* L224 reference materials instead of reference standard
* L230-233 please refer to the figures
* L235 which periods? please clarify

*Results*
* Figure 2: It would be valuable to start at least one month before the starting of the sampling period. Because there is a lag between the timing of particles produced in the ML and when they reach the sediment trap, we are missing the peak of Chl a that corresponds to the material collected in the first cup. Moreover, it will be useful to have the timing of the peaks illustrated (arrows for example) on the figure as well. Please add a legend too.
* Figure 3: Please see my previous comments in the "Major concerns" section.
* Authors mentioned that there is a time lag in the flux of particles between the two traps but not in the stable isotopic composition (e.g. d13C). It will be interesting to discuss the reason of this difference.
* Table 1: It seems that replicate have been made so sd can be calculated and error can be propagated to better represent the error associated with the value presented in this table (for methods to propagate error see for example the Eurachem publication "Quantifying Uncertainty in Analytical Measurement")
*L352 It would be great to mention which cup have been chosen for the microscopic analysis in the method section: Moreover, the deep and shallow cups seem to be from a different period in March 2018. Please explain why and/or any bias associated with this choice or correct the misalignment in the figure.

*Discussion*
* L385 Perhaps specify that the timing fits but not the magnitude of the peaks.
* L388 Please use "additional" instead of "third" since this peak is between the two main peaks
* L390 "PN fluxes followed the same seasonal trend as POC" please develop a little bit more, is it expected? why?
* L409 and after: higher sinking rates could also explain the observations and are consistent with no time lag in 2$^{nd}$ event compared to 1$^{st}$ event
* L415 A figure with simple linear regression would have illustrated this statement.
* L421 "variations" instead of "shifts"
* L422-423 Here too it needs a figure to illustrate this linear regression
* L427 If I'm correct, all diatoms belong to Bacillariophyceae not just Fragilariopsis spp
* L428-430 The low BSi d30Si at the beginning of the bloom is likely explained by a Si source that is already light, rather than more fractionation from diatom such as suggested. Using simple (not perfect) conceptual models such as Rayleigh or Steady-state, one can estimate the isotopic value of this source of Si. Diatoms at the end of summer also fractionate Si isotopes but

they use a Si source that is enriched in 30Si (higher d30Si value). Light d30Si can also come from bias due to the presence of LSi that would be more important early spring (perhaps brought from ice drafts?).

* L473 "significant" instead of "sharp". Using conceptual models and an estimation of the isotopic signature of the Si from deeper water (same as the early spring Si source for example?), one can estimate the amount of Si supplied to the ML (see Fripiat et al 2011 for the methods using seawater samples and Closset et al 2015 for the methods applied to sediment trap samples). Although, there is no chl a in August in figure S2, so the uptake in the surface layer is probably not significant during this period. Additionally, what about LSi contribution to those samples?

* L498-500 In the deepest trap, the error associated to mean value in July is probably too high to conclude anything about a trend in d15N. It could be increasing only from June to August, just as in the shallowest trap. Although I am not against the hypothesis of material coming from different sources and at different stage of degradation, which is very likely during this time of the year.

* L504-505 Unclear

* L527 Please define "isotopic baselines" as it is not explained anywhere in the manuscript. Moreover, there is no baseline shown in figure 3 neither in Table 1. How is this isotopic baseline estimated?

* L539 Another linear regression that needs to be illustrated by a figure (at least in supplementary materials)

* L568 and after: What about the case of a continuous, or semi-continuous supply of DSi to the ML? or the influence of open water vs. sea ice diatoms? Also, sane comment as in L473, the magnitude of Si supplied to the ML can be estimated using simple conceptual models. Also, a BSi d30Si of 0.48 pmil will correspond to a Si source with a d30Si of ~1.68 pmil, which is not too light compared to the d30Si value of Southern Ocean deep water that can be considered as the Si source (e.g. in a quite similar configuration, WW Si source above the Kerguelen plateau has a d30Si of 1.71 pmil, Closset et al 2016)

* L578 Closset et al. 2022 and Cassarino et al. 2020 as reference for pore water d30Si and diffusive Si flux from sediments in the Southern Ocean

* L606 If faecal pellets or moults have been counted it has to be presented in a figure (at least in the supplementary material).

*Conclusion*
The conclusion (and not summary) is generally confusing and need some work to make it clearer. Although I tend to agree with the statements authors are providing, the data presented in this manuscript unfortunately do not support the conclusion (they probably have the potential to do it if better presented and described in the figures)

*References*
Please correct errors in the references (e.g. L791, L797)

---

## Referee Comment (RC2)

Review Biogeosciences Discussion (March 2023)
Seasonal cycles of biogeochemical fluxes in the Scotia Sea, Southern Ocean: A stable Isotope approach
Belcher et al.

The manuscript „Seasonal cycles of biogeochemical fluxes in the Scotia Sea, Southern Ocean: A stable Isotope approach" authored by Belcher et al. investigates the particulate material (and fluxes) from sediment traps in the northern Scotia Sea from different seasons. The data is of high quality and most aspects of the findings are adequately discussed. However, I have some moderate (major) comments (and a few minor, see below) that needs to be addresses. As this topic fits well into the scope of BGD, I recommend publication after careful revision.

**General comment**

I think, this is a great dataset, especially the combination of sediment trap data (even from two different depths and seasons) with three different stable isotope systems. However, some parts of the manuscript are a bit hard to read (and I had to re-read couple of times). It rather reads like a long description of result, whereas, in my opinion, some important aspects are missing. The authors never show or discuss in detail about the POC/PN to BSi ratios. This could shed some light on the connection between the silicon and the carbon cycle and especially the carbon drawdown associated to siliceous phytoplankton. Did the authors ever plotted, the d13C and d30Si data against each other (or d15N versus d15Si). I think it would be really interesting to see, how they positively and partly also negatively (d30Si with d15N) correlate. However, in order to address these issues new figures (e.g. POC/BSi ratios) have to be included in the main text and parts of the result as well as the discussion have to be re-written.

**Methods**

L145: Please check the coordinates, I guess you mean 54.8036°S and 40.1593°W, the "minus" is used for "South" and "West". If you state the direction, you do not have to use the "minus".

**Results**

L352 Did the assemblages only include siliceous plankton (like diatoms and silicoflagellates)? Or did you also observe other taxa (e.g. dinoflagellates, coccos). Maybe you can add one sentence in the beginning that states that only specific type of plankton was observed.

L344 "…, but shallow and deep traps have d15N of similar magnitude". Not sure, if I understand the sentence correct. D15N in shallow samples are much higher compared to d15N in deep sediment trap samples. Even though error is large, the highest mean (shallow) is associated with the lowest (mean). I think, this is an interesting observation, that is not sufficiently discussed. What would be the consequences for paleo reconctructions, if we observe difference in d15N with depth.

Table 1: Can you please be more precise on how the error of the mean is derived. How does it include the analytical as well as the replicate error? Is it a 1 sd error or a propagated error? Please provide more information.

L366 The authors list Dictyocha together with all the diatoms, but it is a silicoflagellates. I think, it would be good, if the taxonomic groups (e.g. diatoms, silicoflagellates, dinoflagellates) are given (see also the comment above).

**Figures**
Figure 3: The figure should be improved. The authors could display the fluxes as boxplots. Please increase the dots size and choose different colors, e.g. open versus filled in black.

Figure 4: It is hard to read the legend and the x, y scale. Could you please increase the font. Can you please specifiy, what the difference between A) and C) and B) and D).
Is this for different seasons. The authors could add an additional legend to make it more clear or edit the figure caption.

**Discussion**

Please note here also my "general comment" in the beginning. I think the manuscript would benefit form a discussion on elemental ratios as well as a comparison between the stable isotopes. Additional figures could emphasize some parts of the discussion (e.g. L415).

L418 Even though POC and BSi can be closely linked and transfer carbon to the deep, the following statement (L418) has to be rephrased. Not all diatoms have greater densities and higher sinking velocities compared to non-siliceous phytoplankton. Sinking velocities are linked to size (e.g. Bauman et al., 2023)
https://egusphere.copernicus.org/preprints/2022/egusphere-2022-814/) . Some fast bloomers (small, e.g.Chaetoceros spp.) often does not sink to the sediment (at least not the vegetative cells), as they are already remineralized in the upper water column. Instead, the big "late bloomers", at the end of a succession (e.g. Coscinodiscus) are often the ones, that are found in the sediments. For a comparison between plankton assemblages in surface water and sediments see also Grasse et al., 2021*.
The authors have information about the biovolume, how does this align with the rest of the data. Maybe they could refer to some of their findings here.

L566 what is the reference for the "exception of one culture study". What exactly is the isotopic baseline, the authors referring to.

* Grasse, P. *et al.* Controls on the Silicon Isotope Composition of Diatoms in the Peruvian Upwelling. *Frontiers Mar Sci* **8**, 697400 (2021).

---

## Author Comment (AC1)

**Response to reviewers: Seasonal cycles of biogeochemical fluxes in the Scotia Sea, Southern Ocean: A stable isotope approach, by Belcher et al.**

**Reviewer 1**
 **General comments**
In "Seasonal cycles of biogeochemical fluxes in the Scotia Sea, Southern Ocean: A stable isotope approach", Belcher et al. present a study investigating the seasonal variations of organic matter (POC and PON) and biogenic silica fluxes from two sediment traps located north-West of South Georgia in the Scotia Sea (Southern Ocean). Using stable isotope approaches the authors examine the origin and some of the processes controlling the fluxes they have observed in the traps.

They investigated the differences between two productive events (in February 2018 – summer season – and December 2018 – spring season) and the coupling of C, N and Si fluxes during these events. Their main results are: Particulate fluxes and isotopic compositions were similar in the deep and shallow trap suggesting that most of the remineralization occurred in the upper layer of the water column. Despite a very noisy d15N signal, the synchronicity if the d30Si and d13C signals highlight the coupling between these two elements and the significant role of diatoms in the export of C (and BSi) in the area. Based on the estimation of isotopic baselines associated with the two productive events, they also suggested a change in the source region of the material coming into the sediment traps.

Generally speaking, the results of this study are interesting. However, I found the manuscript rather hard to read, often unclear or confusing. I think that the manuscript would profit from an effort to make the structure of the discussion more easily understandable for the reader. More work can also be done regarding the description of the analytical and sample processing methods as well as data quality. Some important information is missing and/or unclear. But most importantly, I think the authors should re-think figure 3, which is one of the most (if not the most) important figure of the manuscript. Indeed, the way the figure is built does not support or illustrate the statements or hypothesis authors are attempting to demonstrate. Additionally, there are also some inconsistencies in the wording, and I would strongly encourage the authors to carefully read their manuscript again and have it read by an English-speaking person.

I detail these points below, together with minor points that the authors should also consider while carefully revising this manuscript. I recommend publication of this paper in "Biogeosciences" after major revisions.

**Reply: Thank you for taking the time to review our manuscript and for the helpful suggestions. We believe that we have improved the paper by addressing the points that you have raised. See our responses below. Where we quote line numbers we refer to the marked up version of the revised manuscript.**

**Major concerns**
Currently, the manuscript requires very careful reading (and re-reading) in order to understand the authors' argumentation and get a sense of the various settings. A few suggestions:

* Re-organizing the discussion based on the three main periods that are discussed in the manuscript. For example, having three subsections in chronological order (i) Early spring event (P2), (ii) Late summer event (P1), (iii) Winter hiatus; or to fit with the main figure of the manuscript (fig 3) (i) First export event (P1), (ii) Winter hiatus, (iii) Second export event (P2).
**Reply: Thank you for your suggestions to help improve the readability of the manuscript. We have broken section 4.2 up into 3 sections: productive period 1, winter hiatus and productive period 2**

**as this we think fits well with our figures and other text in the manuscript. We keep the fluxes and isotopic ratios in separate sections to try and keep the isotope section as focussed and clear as possible. With three biogeochemical fluxes/isotopic ratios to describe over an entire season there are many complexities to untangle from our limited dataset and it is not a simple story to tell. We believe we have explained the different processes and hypothesis for our data more clearly in this revised version.**

* As it is, the manuscript needs desperately figures that will support the authors' hypothesis and statements for two main reasons: (i) Some important figures are missing. For example, figures illustrating the relationship between POC and BSi (mentioned for example L415) or d13C and d30Si (mentioned L422) with associated R2 and statistics. Right now, there is no figures to illustrate or support the relationship authors are discussing in the manuscript.ii) Figure 3, the key figure of the manuscript, is currently very poorly designed. The choice of shading to represent fluxes in mg m-2 d-1 does not actually reflect the full magnitude of those fluxes. The most obvious example is POC flux in the shallow trap in May 2018. It seems to "peak" for a short period of time to values around 8 mg m-2 d-1 while it was sustaining this rate for a long period of time (31 days). Fluxes will appear less biased by using barplot representing the mean flux and error bar for the standard deviation. The variations of the isotopic composition of particles (d13C, d15N and D30Si) are also poorly illustrated by the choice of representing only the min and max on the figure. A mean value with error bar will be more representative of the seasonal evolution of the signal, as well as of the heterogeneity of the material (when error bars are more widely spread). It will also help with the scattering of the d15N and validate (or invalidate) the trends suggested by the authors.

* **Reply: We have added an additional figure (Figure 5) with the linear regressions suggested (see later response), and additionally we have changed figure 3 to have bars as well as a shadowed region to show the maximum and minimum. This allows us to display the shallow and deep sediment trap data on the same plot without obscuring values with the use of error bars, whilst still being clear about the range of values for the replicate splits analysed. The bars have a width proportional to the time period that the cup was open for clarity. For the isotope ratios we keep the use of the error bar in the bottom right-hand corner of each plot. This maintains clarity on the plot, preventing it from being too cluttered and illegible were there to be uncertainty shown for each individual point.**

[Figure]

[Figure]

[Figure]

\* Something that need to be highlighted in the method section and briefly discussed later on is that the total collection period is 341 days. The first cup opened on Jan 25th (2018) and the last one closed on Jan 1st (2019). I would be worth it mentioning that the sediment trap series misses 3 weeks in the beginning of January where the flux is expected to be significant. Authors have not made any annual/seasonal integration of their fluxes, but they should still discuss the risk of missing a significant part of the seasonal flux early in the season. It might be of importance for the discussion regarding the isotope baseline for the first export event.

**Reply: We have added in the deployment dates in the methods, and also state on lines 311, where we define productive period 1, that we do not capture the first 3 weeks of january. In the discussion we say that the record does not extend beyond December 2018 so cannot determine if the isotope ratios would return to values akin to that of period 1. We remind the reader that we do not capture the first 3 weeks of January in the discussion (Lines 469 and 685).**

\* It is not clear in the method section if authors have used the different splits of samples as replicates or if they have combined splits to do their different measurements. Figure 3 gives two values for each measurement (a min and a max) so one can guess that authors have measured duplicates out of those splits. Going through the discussion section, authors start to mention these mean values that do not correspond to anything presented in figure 3. I do not see the point of plotting only the min and max on the figure while using a mean value in the text. This is confusing, make things unclear and prevent the figure to illustrate and support correctly the text (e.g. it is hard to see some of the trends that are discussed in the text). I am suggesting using the mean values in the figure with the corresponding error bar and add those error (as standard deviation) within the text.

**Reply: As noted below we have tweaked the text to clarify that for each sediment trap sample bottle, splits were taken for the different analyses, and typically two splits were used for each analysis – thus giving two replicates. The values of these are plotted in the figure as the maximum and minimum. In the text, we feel it is most useful to give the mean of our sediment trap sample replicates (from multiple splits of the same sediment trap sample bottle), since this keeps the text concise. To avoid confusion, we define at the start of the results that we are referring to the mean result for each sediment trap bottle based on available split samples, and remove the use of the word mean in these instances. We add in the following lines (320-322) to explicitly state this at the start of the results section.**

*"Since two to three splits were analysed from each sediment trap bottle, we refer here to the mean flux for each sediment trap bottle based on the available splits for that bottle."*

**Since typically we only have two splits per sample, we believe the range to be an informative way of illustrating the spread of our results. In the new bar plot for figure 3 we plot the mean flux value and use shading to show the maximum and minimum flux value.**

\* Please define what "isotopic baselines" is. It is not defined anywhere in the manuscript, neither it is explained how authors have estimated the different baseline they are referring to. If it is just the lightest isotopic signature recorded just before a productive (and export) event, I am suspicious with the isotopic baseline identified for the first event since a significant part of the flux might have been missed early January 2018. Moreover, the isotopic baselines are not identified in figure 3 neither in Table 1. In general, this last part of the discussion (related to the comparison of the different isotopic baselines) is quite confusing and unclear and need some serious rephrasing.

**Reply: To avoid confusion we have removed all mention of the term isotopic baseline in the manuscript**

**Minor concerns**
*Introduction*
\* L43 Use biological pump of carbon instead of BCP.

**Reply: Here and throughout we remove the acronym BCP and use the standard terminology in the literature of 'the biological carbon pump'.**

* Sediment traps have bias too. A short summary (one-two sentences + ref) of them would be useful here. Especially since authors discuss some of them later in the manuscript.
**Reply: Added in the following lines:**
*"Sediment traps can be susceptible to collection biases depending on the depth of deployment, trap design, hydrodynamic conditions and properties of sinking particles (Buesseler et al., 2007). Moored sediment traps can underestimate the actual flux at depths shallower than ~1500 m by collecting only a portion of the sinking material, though biases vary greatly between sites (Buesseler et al., 2007)."*

* L73 Use "challenges" instead of "complications"
**Reply: Changed as requested**

*Does sea-ice affect the region where the traps are located? It has been shown that the occurrence of sea ice can significantly affect stable isotopes composition (at least for d13C - e.g. Kennedy et al. 2002 - and d30Si - e.g. Fripiat et al. 2007). This could be an important aspect to consider in your discussion.
**Reply: Sea ice does not reach this region, thus we do not discuss it.**

*The processes controlling the stable isotopic composition of C and N are quite well introduced but authors have been very quick concerning d30Si. More information about fractionation and the processes controlling it need to be introduced here (e.g. difference in fractionation between polar diatom species - Sutton et al. 2013 – fractionation (or not) associated with biogenic silica dissolution - Demarest et al. 2009, Wetzel et al. 2014).
**Reply: We have added the following additional text here (lines 97-110)**
*"During uptake of DSi, diatoms fractionate the stable isotopes of silicon (28Si, 29Si, 30Si) preferentially taking up the lighter isotopes during cell wall (frustule) formation (De La Rocha et al., 1997). This means that BSi fluxes and ratios of light 28Si to heavy 30Si (expressed as δ30Si) in sinking particulate organic matter (POM) can be informative about DSi utilisation by siliceous phytoplankton. The fractionation of Si isotopes during diatom DSi utilisation is approximately -1.1 ‰, although estimates of this value vary in laboratory and field studies between -0.5 and -2. 5‰ (Hendry and Brzezinski, 2014). Whilst some studies have shown that isotopic fractionation is independent of temperature, DSi and diatom species (e.g., De La Rocha et al., 1997), one in vitro laboratory culture experiment revealed a potential species effect, with polar species exhibiting more extreme fractionation (-2.09 ‰for Chaetoceros sp. and 0.54 ‰ Fragilariopsis kerguelensis, Sutton et al., 2013). The impact of water column dissolution on frustule δ30Si is poorly constrained, with experimental evidence for either a small fractionation of -0.55 ‰ (Demarest et al., 2009) or a negligible impact (Wetzel et al., 2014; Egan et al., 2012; Grasse et al., 2021)."*

*L115-116 what is the difference between annually and seasonally? Are they calculated over a different period?
**Reply: Wording amended to clarify (line 133-136):**
*"Faecal pellets (up to 91 % in late spring and early summer seasonally, Manno et al., 2015), krill exuviae (up to 47 % in summer, Manno et al., 2020) and diatoms, particularly resting spores (annual contribution of 42 %, Rembauville et al., 2016)…"*

* Do the stable isotopes really give information about the actual composition of organic matter?
**Reply: rephrased as follows (lines 137-142):**

*"Here we use $\delta^{13}C_{POC}$, $\delta^{15}N_{PN}$ and $\delta^{30}Si_{BSi}$ alongside calculated fluxes of POC, PN and BSi as tools to reveal information about sinking particulate organic matter and the processes influencing its production and subsequent flux to depth. More in-depth understanding of the composition, and thus the drivers of POC flux in this important region are key to improving estimates of the current and future strength of the biological carbon pump and the ocean's role as a CO2 sink."*

\* Perhaps what is missing in the introduction is a paragraph about why is the composition of particles important for the biological pump of carbon? For example, it will affect the sinking speed of particles (and authors discuss this later), their recycling in the water column (and later within the sediment) etc.
**Reply: Added following line in opening paragraph of introduction (lines 50-52):**
*"The composition of particles affects the sinking rate, lability and thus degree of remineralisation as they sink through the water column (e.g. Ploug et al., 2008; Giering et al., 2020)."*

*Material and Methods*
\* L152-153 it would be interesting to quantify this effect in percent of the signal
**Reply: Added in the following text (lines 172-175).**
"Previous studies have reported the effects of formalin on $\delta^{13}C_{POC}$ and $\delta^{15}N_{PN}$ to be small (±1 ‰ and ±1.5 ‰ respectively, Mincks et al., 2008 and references therein). This equates to 13 % and 16 % of the maximum range measured in our study, which is small compared to the isotopic shifts we observed."

\* L168 splitted
**Reply: The use of split here is correct, thus we do not change it.**

\* It is not clear in the different paragraphs of the method section if these slips were combined or analyzed separately as replicates. If it is the second option (which is my guess) these replicates can be used to calculate the error or std on the samples as the potential heterogeneity of the sample will be reflected by a large error bar on the sample.
**Reply: Splits were taken from one sediment trap sample and analysed separately as replicates. This is the range we use in the shading in figure 3 since, as you say, the sample heterogeneity brings a large range. Text has been tweaked in the methods to clear this up. We have altered figure 3 as suggested, and the range (max to min) is shown by shading to highlight the uncertainties.**

\* L183 "per mil" instead of "per mille"
**Reply: per mille is correct so we have kept this**

\* Please define the meaning of "PACS international standard"
**Reply: We have added the full description: PACS-2 marine sediment reference material.**

\* L191 several statistical errors have been described here, although it is mentioned later in the manuscript, authors should indicate which one they have associated to their measurement).
**Reply: We state in the methods the likely uncertainty surrounding the formalin preservative on C and N isotope ratios, stating that this is larger than analytical uncertainties. We state this in both section 2.2 and 2.3.1. We represent these uncertainties on the isotopic ratios visually on figure 3 with the error bar in the bottom right corner, as detailed in the caption. As explained later we use pooled variances to quantify the uncertainties on the calculation of the seasonal flux-weighted isotopic ratio.**

\* It would be interesting to quantify those splits in percent of the total sample

**Reply: Splits ranged from 0.008 of a sample to 0.12 of a sample. This is based on the quantity of material collected in the cup and the amount needed for analysis of the different isotopes/fluxes. We do not think this information adds any value to the analysis so do not add this in to keep the manuscript as concise and clear as possible.**

\* What about lithogenic silica? Alkaline extraction method using NaOH will dissolve some lithogenic material along with BSi (Ragueneau et al. 2005). Because LSi has a light d30Si (down to -2.3 pmil, Opfergelt and Delmelle 2012), it has the potential to bias BSi d30Si measurements even with low LSi contribution (or contamination) to the alkaline digestion. For example, and in the worst-case scenario of LSi d30Si of -2.3 pmil, a contribution of 3% during period 2 and 4% during period 1 would significantly bias the result (by 0.1pmil). This is a rough calculation, but this need to be discussed as sediment traps can collect a significant amount of lithogenic material. Nota that methods have been proposed to "correct" BSi d30Si from LSi contamination (see Closset et al 2015).

**Reply: Many thanks for this suggestion. The reviewer is correct that there is potential for lithogenic contamination with the method that was used. However, we do not have the required Al/Si data from the samples due to sample limitation, and do not have robust information about the Al/Si content of the potential endmembers in this region, and so have decided against introducing uncertainty by attempting a lithogenic correction. As noted by the reviewer, the potential bias from such lithogenic contamination is an order of magnitude smaller (c. 0.1‰) compared to the signal observed (over 1‰). Furthermore, the excellent reproducibility between replicates suggests against contamination, which is unlikely to be consistent from sample to sample.**

**To address this, we have added the following to the text (lines 259-262):**

*"A lithogenic correction (e.g., Closset et al., 2015) was not carried out on these samples. However, even an extreme scenario of variable lithogenic contamination of 1-5% of isotopically light marine clays (with $\delta^{30}Si$ of -2.3‰; Opfergelt and Delmelle, 2012) would only result in a potential systematic offset of 0.12‰, which is an order of magnitude smaller than the observed seasonal signal."*

\* L206 HCL or Milli-Q water as eluent?

**Reply: Wording altered here for clarification**

*"For Si isotope analysis, supernatants and reference standards were purified by passing through cation exchange columns (Bio-Rad AG50W-X12, 200-400 mesh resin) pre-cleaned with HCl following Georg et al. (2006)."*

\* L222 This sentence is unclear. what are those pseudo replicates? two samples per pseudo replicates so four isotopic measurement per bottle? Isn't the case for all isotopic measurements (and for POC and PON fluxes too)?

**Reply: we have deleted the term pseudo replicates for clarity. Two or three splits (depending on material availability and analysis possible) were taken from each sediment trap and analysed separately. This was the case for all of the different measurements. We have tweaked the wording to make this clearer.**

\* L224 reference materials instead of reference standard

**Reply: Done**

\* L230-233 please refer to the figures

**Reply: Done**

\* L235 which periods? please clarify

**Reply: Done**

*Results*

* Figure 2: It would be valuable to start at least one month before the starting of the sampling period. Because there is a lag between the timing of particles produced in the ML and when they reach the sediment trap, we are missing the peak of Chl a that corresponds to the material collected in the first cup. Moreover, it will be useful to have the timing of the peaks illustrated (arrows for example) on the figure as well. Please add a legend too.

**Reply: Thank you for the helpful suggestions , we have made these changes**

* Figure 3: Please see my previous comments in the "Major concerns" section.

**Reply: Amended as detailed above**

* Authors mentioned that there is a time lag in the flux of particles between the two traps but not in the stable isotopic composition (e.g. d13C). It will be interesting to discuss the reason of this difference.

**Reply: For both isotope ratios and fluxes, whether we see a lag or not depends on the balance between the sampling resolution and the abruptness in the change of the magnitude and composition of the sinking material. There is a time lag between the flux of particles reaching the deep and shallow traps, in that the majority of particles reach the deep sediment trap later due to the time taken to sink deeper. However, the start of the productive flux events is captured in both traps with no lag, highlighting fast sinking material (at least within the temporal resolution of the sampling cup). We discuss in lines 347-354 the possible inferences about sinking velocity that can be made by examining the lags between fluxes. Considering the low background flux of material over winter, only a small amount of additional material would be needed to impact the stable isotope ratios, thus we do not see the time lag here. There is likely some lag in the isotope ratios, but not that we can capture in the temporal resolution of the sediment traps.**

* Table 1: It seems that replicate have been made so sd can be calculated and error can be propagated to better represent the error associated with the value presented in this table (for methods to propagate error see for example the Eurachem publication "Quantifying Uncertainty in Analytical Measurement")

**Reply: Considering the small number of repeat measurements, we chose to use pooled variances as a measure of uncertainty and have used these in table 1 in place of the uncertainties surrounding the formalin preservative. Where the number of degrees of freedom is small (i.e. with 2 repeat measurements, as in our case), the 95% confidence interval for the standard deviation is 0.03-2.24 σ, which is a range that spans a factor of almost 100. Pooling 10 such twice measured samples gives a 95% confidence of the sample interval of 0.57-1.4 σ. Pooling the variance of measurements (i.e. from each sediment trap cup measured) made within the time period of interest (here, full season, period 1 and period 2, DOF = X, Y and Z respectively), increases the robustness of the uncertainty estimate. This method assumes that the underlying distributions from which the data are drawn all have the same scatter. Table 1 has been appended with these pooled variances, and the caption amended to state that we use pooled variance as a measure of uncertainty on the seasonal flux-weighted isotopic ratios.**

*L352 It would be great to mention which cup have been chosen for the microscopic analysis in the method section: Moreover, the deep and shallow cups seem to be from a different period in March 2018. Please explain why and/or any bias associated with this choice or correct the misalignment in the figure.

**We add in a reference to table 2 where the cup timings for these samples are given. Samples were chosen to capture the peak fluxes in each sediment trap. The peaks do not align exactly in period 1 for deep and shallow traps, hence differed periods were chosen, this information is given in the text (lines 414-415):**

*"Eight samples (four deep and four shallow, table 2) were analysed by light microscope for phytoplankton composition to cover the high productivity periods 1 and 2."*

*Discussion*

\* L385 Perhaps specify that the timing fits but not the magnitude of the peaks.

**Reply: Our text explains this, as follows.**

*"The seasonal cycles of POC agree well with previously published work at the same location (Manno et al., 2015), with peaks in austral spring and late summer, though the peak POC fluxes recorded here (means of 45.7 mg C m$^{-2}$ d$^{-1}$ and 43.4 mg C m$^{-2}$ d$^{-1}$, in shallow and deep traps respectively) are higher than those observed in previous years (22.9 mg C m$^{-2}$ d$^{-1}$; Manno et al., 2015)."*

\* L388 Please use "additional" instead of "third" since this peak is between the two main peaks

**Reply: Done**

\* L390 "PN fluxes followed the same seasonal trend as POC" please develop a little bit more, is it expected? why?

**Reply: We have added the following text in the discussion (lines 455-460):**

*"PN fluxes followed the same seasonal trend as POC for both deep and shallow traps suggesting a similar source. The similar magnitude of POC:PN ratios in period 1 in the two traps support consistency in the degree of degradation at these depths. The lower POC:PN ratios measured in the deep trap between August and October, compared to the shallow trap are consistent with a divergence in δ15NPN ratios, and could relate to a change in source material and/or degradation state between the two traps at this time."*

\* L409 and after: higher sinking rates could also explain the observations and are consistent with no time lag in 2nd event compared to 1st event

**Reply: we have added sinking rates in more explicitly as another explanation (see line 484).**

\* L415 A figure with simple linear regression would have illustrated this statement.

**Reply: We have added in the following figure (figure 5).**

[Figure]

* L421 "variations" instead of "shifts"
**Reply: Done**

* L422-423 Here too it needs a figure to illustrate this linear regression
**Reply: Added, as above**

* L427 If I'm correct, all diatoms belong to Bacillariophyceae not just Fragilariopsis spp
**Reply: yes, thank you for spotting this. We have changed the wording to:**
*"This is consistent with the dominance of diatoms (Fragilariopsis spp.) in the trap material"*

* L428-430 The low BSi d30Si at the beginning of the bloom is likely explained by a Si source that is already light, rather than more fractionation from diatom such as suggested. Using simple (not perfect) conceptual models such as Rayleigh or Steady-state, one can estimate the isotopic value of this source of Si. Diatoms at the end of summer also fractionate Si isotopes but they use a Si source that is enriched in 30Si (higher d30Si value). Light d30Si can also come from bias due to the presence of LSi that would be more important early spring (perhaps brought from ice drafts?).

**Reply: Many thanks for this comment. Please see response to the comment below.**

\* L473 "significant" instead of "sharp". Using conceptual models and an estimation of the isotopic signature of the Si from deeper water (same as the early spring Si source for example?), one can estimate the amount of Si supplied to the ML (see Fripiat et al 2011 for the methods using seawater samples and Closset et al 2015 for the methods applied to sediment trap samples). Although, there is no chl a in August in figure S2, so the uptake in the surface layer is probably not significant during this period. Additionally, what about LSi contribution to those samples?
**Reply: Many thanks for these suggestions. We had considered including such a conceptual model. However, as the reviewer notes, these models are highly conceptual and we consider them – in our case – to be under-constrained and unlikely to shed additional light on the interpretation. In addition, given the balance of the paper between the C, N, and Si results, if we were to model the Si component of our findings then it would also be logical to model the C and N components. Such modelling efforts would be out of the scope of the current study. As such, we respectfully suggest that this modelling work is not required for this manuscript. Please also see our comment above regarding the lithogenic contamination.**

\* L498-500 In the deepest trap, the error associated to mean value in July is probably too high to conclude anything about a trend in d15N. It could be increasing only from June to August, just as in the shallowest trap. Although I am not against the hypothesis of material coming from different sources and at different stage of degradation, which is very likely during this time of the year.
**Reply: We have removed some text from here and edited the wording so we do not overstate any trend in the deep d15N at this time.**

\* L504-505 Unclear
**Reply: We have ammended these lines to make them clearer**
\* L527 Please define "isotopic baselines" as it is not explained anywhere in the manuscript. Moreover, there is no baseline shown in figure 3 neither in Table 1. How is this isotopic baseline estimated?
**Reply: To avoid confusion, we have removed the use of the word baseline in this context from the manuscript.**
\* L539 Another linear regression that needs to be illustrated by a figure (at least in supplementary materials)
**Reply: We feel that the point we are making is sufficiently evidenced by statistics and do not add the figure requested as this will clutter the manuscript to add 4 regression plots.**

\* L568 and after: What about the case of a continuous, or semi-continuous supply of DSi to the ML? or the influence of open water vs. sea ice diatoms? Also, sane comment as in L473, the magnitude of Si supplied to the ML can be estimated using simple conceptual models. Also, a BSi d30Si of 0.48 pmil will correspond to a Si source with a d30Si of ~1.68 pmil, which is not too light compared to the d30Si value of Southern Ocean deep water that can be considered as the Si source (e.g. in a quite similar configuration, WW Si source above the Kerguelen plateau has a d30Si of 1.71 pmil, Closset et al 2016)
**Reply: Please see comment above regarding the conceptual modelling. Also, please note that the study area does not experience sea ice, and so sea-ice diatoms are unlikely to be present.**

\* L578 Closset et al. 2022 and Cassarino et al. 2020 as reference for pore water d30Si and diffusive Si flux from sediments in the Southern Ocean
**Reply: Added**
\* L606 If faecal pellets or moults have been counted it has to be presented in a figure (at least in the supplementary material).

**Reply: These were not counted**

*Conclusion*
The conclusion (and not summary) is generally confusing and need some work to make it clearer. Although I tend to agree with the statements authors are providing, the data presented in this manuscript unfortunately do not support the conclusion (they probably have the potential to do it if better presented and described in the figures)

**Reply: We have edited the conclusions, and shortened the section to remove statements that are not supported by the manuscript.**

*"The seasonal cycles in primary productivity and nutrient uptake in surface waters at our study site in the Scotia Sea are reflected in the fluxes and isotopic ratios of sinking particulate material. We find that most remineralisation occurs in the upper 400 m of the water column and below this the magnitude of the flux of sinking material is relatively consistent, supported by consistency in POC:PON ratios. We find that particulate fluxes of C, and BSi are tightly coupled which highlights the importance of siliceous material in the transfer of POC to depth. We suggest that a change in phytoplankton community structure can at least part explain the shifts in carbon isotopic composition between the two productive periods measured here. Though complex, seasonal patterns in isotopic composition of particulate material reaching the sediment traps do reflect the degree and type of nutrient utilisation in the source waters. Our data also suggests an importance of laterally supplied material to the sediment traps and supports seasonal differences in source regions. Our results highlight how, through more detailed mechanistic understanding of the drivers of POC flux, and biogeochemical cycling, we can improve estimates of the current and future strength of the biological carbon pump and the ocean's role as a CO2 sink."*

*References*
Please correct errors in the references (e.g. L791, L797)

**Reply: Added page number and volume number**

---

## Author Comment (AC2)

**Response to reviewers: Seasonal cycles of biogeochemical fluxes in the Scotia Sea, Southern Ocean: A stable isotope approach, by Belcher et al.**

**Reviewer 2**

The manuscript „Seasonal cycles of biogeochemical fluxes in the Scotia Sea, Southern Ocean: A stable Isotope approach" authored by Belcher et al. investigates the particulate material (and fluxes) from sediment traps in the northern Scotia Sea from different seasons. The data is of high quality and most aspects of the findings are adequately discussed. However, I have some moderate (major) comments (and a few minor, see below) that needs to be addresses. As this topic fits well into the scope of BGD, I recommend publication after careful revision.

**Reply: We thank reviewer 2 for their insights and believe we have made improvements to the manuscript by addressing them below. Where we give line numbers we refer to the marked up version of the revised manuscript.**

**General comment**

I think, this is a great dataset, especially the combination of sediment trap data (even from two different depths and seasons) with three different stable isotope systems. However, some parts of the manuscript are a bit hard to read (and I had to re-read couple of times). It rather reads like a long description of result, whereas, in my opinion, some important aspects are missing. The authors never show or discuss in detail about the POC/PN to BSi ratios. This could shed some light on the connection between the silicon and the carbon cycle and especially the carbon drawdown associated to siliceous phytoplankton. Did the authors ever plotted, the d13C and d30Si data against each other (or d15N versus d15Si). I think it would be really interesting to see, how they positively and partly also negatively (d30Si with d15N) correlate. However, in order to address these issues new figures (e.g. POC/BSi ratios) have to be included in the main text and parts of the result as well as the discussion have to be re-written.

**Reply: We have added in figure 5 (shown below) showing relationships between Si and POC fluxes as well as isotope ratios. Additionally, we have added in information about POC:PN and BSi:POC ratios in the results, and bring this into the discussion. No correlation between Si and N, or C and N isotopes was found based on linear regression analysis. We have revised and restructured the discussion to make the manuscript clearer. In the results section we add (lines 330-335):**

"The mean POC:PN ratio (mol:mol) throughout the study period was 6.40 (± 0.73) and 6.02 (±0.90) in shallow and deep traps respectively with higher ratios in the productive periods compared to the winter months. Mean POC:PN ratios were 6.75 (±0.46) and 6.63 (±0.71) during period 1 and period 2 in the shallow trap, and 6.61 (±0.65) and 5.51 (±0.87) in the deep trap. Over the winter months POC:PN was 5.68 and 5.92 in shallow and deep traps respectively."

[Figure]

**Methods**

L145: Please check the coordinates, I guess you mean 54.8036°S and 40.1593°W, the "minus" is used for "South" and "West". If you state the direction, you do not have to use the "minus".

Reply: Amended

**Results**

 L352 Did the assemblages only include siliceous plankton (like diatoms and silicoflagellates)? Or did you also observe other taxa (e.g. dinoflagellates, coccos). Maybe you can add one sentence in the beginning that states that only specific type of plankton was observed.

Reply: Diatoms, silicoflagellates and dinoflagellates were observed in the sediment traps, we have added a sentence to state this. We also observed micro-zooplankton in small numbers, in particular radiolarian and tintinnids and have added this in (lines 415-418).

*"Diatoms, silicoflagellates and dinoflagellates were observed, with a dominance of diatoms. Micro-zooplankton were also recorded, in particular radiolarian and tintinnids, though these were not dominant by biovolume or abundance."*

L344 "…, but shallow and deep traps have d15N of similar magnitude". Not sure, if I understand the sentence correct. D15N in shallow samples are much higher compared to d15N in deep sediment trap samples. Even though error is large, the highest mean (shallow) is associated with the lowest (mean). I think, this is an interesting observation, that is not sufficiently discussed. What would be the consequences for paleo reconctructions, if we observe difference in d15N with depth.

Reply: We have removed the statement about similar magnitude. We point the reviewer to the discussion, section 4.2.3, lines 644-669, where we discuss more about changes in D15N with depth.

Table 1: Can you please be more precise on how the error of the mean is derived. How does it include the analytical as well as the replicate error? Is it a 1 sd error or a propagated error? Please provide more information.

Reply: Considering the small number of repeat measurements, we chose to use pooled variances as a measure of uncertainty and have used these in table 1 in place of the uncertainties surrounding the formalin preservative. Where the number of degrees of freedom is small (i.e. with 2 repeat measurements, as in our case), the 95% confidence interval for the standard deviation is 0.03-2.24 σ, which is a range that spans a factor of almost 100. Pooling 10 such twice measured samples gives a 95% confidence of the sample interval of 0.57-1.4 σ. Pooling the variance of measurements (i.e. from each sediment trap cup measured) made within the time period of interest (here, full season, period 1 and period 2, DOF = X, Y and Z respectively), increases the robustness of the uncertainty estimate. This method assumes that the underlying distributions from which the data are drawn all have the same scatter. Table 1 has been appended with these pooled variances, and the caption amended to state that we use pooled variance as a measure of uncertainty on the seasonal flux-weighted isotopic ratios.

L366 The authors list Dictyocha together with all the diatoms, but it is a silicoflagellates. I think, it would be good, if the taxonomic groups (e.g. diatoms, silicoflagellates, dinoflagellates) are given (see also the comment above).

Reply: We have added in the taxonomic groups so it is clear which group the species mentioned belong to.

**Figure**

Figure 3: The figure should be improved. The authors could display the fluxes as boxplots. Please increase the dots size and choose different colors, e.g. open versus filled in black.

Reply: We have altered this plot to use bars to better show the length of time each sediment trap sample cup is open for. As requested we have increased the size of the dots, and used open and filled. We keep the blue and red colours for the dots to be consistent with the colours used for the bar plot for deep and shallow particle fluxes.

[Figure]

[Figure]

C) BSi

Figure 4: It is hard to read the legend and the x, y scale. Could you please increase the font. Can you please specifiy, what the difference between A) and C) and B) and D). Is this for different seasons. The authors could add an additional legend to make it more clear or edit the figure caption.

**Reply: We have increased the font size, and also have reworded the caption to make it clearer that the plots show biovolume and abundance**

**Discussion**

Please note here also my "general comment" in the beginning. I think the manuscript would benefit form a discussion on elemental ratios as well as a comparison between the stable isotopes. Additional figures could emphasize some parts of the discussion (e.g. L415).

**Reply: As above, we have added in an additional figure and text.**

L418 Even though POC and BSi can be closely linked and transfer carbon to the deep, the following statement (L418) has to be rephrased. Not all diatoms have greater densities and higher sinking velocities compared to non-siliceous phytoplankton. Sinking velocities are linked to size (e.g. Bauman et al., 2023) https://egusphere.copernicus.org/preprints/2022/egusphere-2022-814/) .Some fast bloomers (small, e.g. Chaetoceros spp.) often does not sink to the sediment (at least not the vegetative cells), as they are already remineralized in the upper water column. Instead, the big "late bloomers", at the end of a succession (e.g. Coscinodiscus) are often the ones, that are found in the sediments. For a comparison between plankton assemblages in surface water and sediments see also Grasse et al., 2021*. The authors have information about the biovolume, how does this align with the rest of the data. Maybe they could refer to some of their findings here.

**Reply: Yes, good point, thank you for spotting this. We have amended the wording as follows:**

*"…suggest an important role of diatoms in transferring organic carbon to the deep ocean at this time. This could be achieved if cells are large through large mineral (silica) ballasted cells sinking at high velocities (Baumann et al., 2022), or through the bioprotection of internal organic matter from grazing and oxidation by the diatom silica frustules (Passow and De La Rocha, 2006; Armstrong et al., 2001; Smetacek et al., 2004)."*

**Since we only have data on biovolume for a small portion of the sampling period, and due to the ability to only count intact cells as explained, we do not think it appropriate to go into more depth in the biovolume data.**

L566 what is the reference for the "exception of one culture study". What exactly is the isotopic baseline, the authors referring to.

**Reply: We have amended the wording here to clarify our meaning.**

\* Grasse, P. et al. Controls on the Silicon Isotope Composition of Diatoms in the Peruvian Upwelling. Frontiers Mar Sci 8, 697400 (2021).

---

## Referee Report (RR1)

**Review Biogeosciences Discussion (June 2023)/2nd round**
**Seasonal cycles of biogeochemical fluxes in the Scotia Sea, Southern Ocean: A stable Isotope approach**
**Belcher et al.**

The revised version is significantly improved. However, I still have some minor comments, that needs to be addressed before publication.

**L205**: The authors give the delta notation only for carbon and nitrogen isotopes, but not for silicon isotopes. I see that the silicon method description is given in another paragraph, but the authors could at least refer to the delta notation, as it is the same for all three stable isotopes (C, N, Si).

**L393:** Other organisms were observed, but not counted? Do the authors have a rough idea of how much of the sediment trap material was diatoms compared to other (non-siliceous) organisms)? This has some implications for the interpretation of $\delta^{13}C$ and $\delta^{15}N$. See also my comment below (L.474). In line 395, the authors say "with a dominance of diatoms". What exactly does that mean? More than 50% or 90%? The following paragraph only gives information about the diatoms and a few silicoflagellates in each sample. How many dinoflagellates do the authors observe? Does "other" in Figure 4 refers to other taxa, like dinoflagellates? Or other diatoms? Please clarify in the text and the figure caption.

**L429:** It is very interesting to see the comparison to other flux measurements in the region. However, I miss some kind of interpretation here. Why are POC and BSi fluxes generally higher, but much lower compared to Closset et al. (2015)? Any major changes in the area, that are causing this. Why does the sampling location from Closset et al. (2015) have more than 10x higher fluxes compared to this study?

**L469:** Please check the sentence. Something is odd. "This could be achieved if cells are large through large".

**L474:** I think, this is not even a "broadly" similar trend in Figure 5b. I think the authors should rather discuss, why they do not see a linear trend between $\delta^{13}C_{POC}$ and $\delta^{30}Si_{BSi}$. Even though the particulate ratios show a strong relation between POC and BSi (except for 3-4 points above the line), the less pronounced or not present relationship in the isotopes can have several reasons.

1. more variation and a higher range in the fractionation factor for $\delta^{13}C$ compared to $\delta^{30}Si$ (e.g. Brandenburg et al., 2022[1]), which can also include different trophic levels.
* * *
[1] Brandenburg, K. M., Rost, B., Waal, D. B. V. de, Hoins, M. & Sluijs, A. Physiological control on carbon isotope fractionation in marine phytoplankton. *Biogeosciences* **19**, 3305–3315 (2022).

2. non-siliceous organisms or organic material (dinoflagellates, microzooplankton). Whereas $\delta^{30}$Si is measured mainly in diatoms, $\delta^{13}$C, as well as $\delta^{15}$N, is measured in other materials/organisms as well.
3. different remineralization for organic carbon and silicon in the frustule

**L476:** The authors state that they do not find significant relationships between $\delta^{13}$C$_{PON}$, $\delta^{13}$C$_{POC,}$ and $\delta^{30}$Si$_{BSi}$. It would be good if the authors could either show the figures in the supplement or report the $r^2$ and p levels here for comparison. I am a bit surprised, that the relationship in Figure 5b is significant. Did you include the error? Please check again.

**L487**: This is more of a general comment. Do the authors take the sinking velocity of particles into account, when discussing the sediment trap data? And if yes, what is the sinking velocity they assume?

**L493:** "BSi: POC ratios were elevated at the start of productive period 1, suggesting that phytoplankton were heavily silicified…..this statement can only be made if the ratio of siliceous to non-non-siliceous plankton is not changing over time. Here the authors need to give more information about the amount of dinoflagellates in their samples (see also statement above). And if the statement is "true", why should a more intense silification is observed?

**Figure 3:** The figure did improve significantly. Maybe it is possible to additionally add the legend to the figure for the deep (red) and shallow (blue) sediment traps.

**Figure 4:** Maybe the authors could either highlight it in the figure or in the figure caption, how does the assemblage data fit to their different time periods? The abundance data at the end of the sampling campaign (Dec./Jan.) fit within the productive period 2, but the first was already in the winter hiatus.

**Figure 5:** Please add error bars for the isotope data in 5b.

---

## Referee Report (RR2)

**General comments**

In "Seasonal cycles of biogeochemical fluxes in the Scotia Sea, Southern Ocean: A stable isotope approach", Belcher et al. present a study investigating the seasonal variations of organic matter (POC and PON) and biogenic silica fluxes from two sediment traps located north-West of South Georgia in the Scotia Sea (Southern Ocean). Using stable isotope approaches the authors examine the origin and some of the processes controlling the fluxes they have observed in the traps.

They investigated the differences between two productive events (in February 2018 – summer season – and December 2018 – spring season) and the coupling of C, N and Si fluxes during these events. Their main results are: Particulate fluxes and isotopic compositions were similar in the deep and shallow trap suggesting that most of the remineralization occurred in the upper layer of the water column. Despite a very noisy d15N signal, the synchronicity if the d30Si and d13C signals highlight the coupling between these two elements and the significant role of diatoms in the export of C (and BSi) in the area. Based on the estimation of isotopic baselines associated with the two productive events, they also suggested a change in the source region of the material coming into the sediment traps.

Having reviewed the first version of this manuscript, I greatly appreciate authors' efforts to improve the reading by carefully re-organizing the different sections. The introduction is clear and describes all the background needed to fully appreciate the manuscript. New elements have been added to the discussion and greatly improve the manuscript. Some minor points will still benefit to be clarified and, although they have already greatly improved the figure, I am personally not convinced that figure 3, which is the most important figure in the manuscript, is not presented in the clearest/smartest possible way (but this is my personal taste, the data are currently there).

I detail these points below and, although I recommend publication of the manuscript after minor revisions, I am convinced that this paper will be a great addition to "Biogeosciences".

**Methods**
* L238: Pioneer ref is Cardinal et al. 2003
* L249-253: A 0.12‰ offset in $\delta^{30}Si$ value might be an order of magnitude lower compared to the seasonal signal, however it is significant regarding the magnitude of $\delta^{30}Si$ variations measured in the study (and could potentially be higher than the error calculated on duplicates or using standards). I am aware that this offset is the wort case scenario, and I am convinced about the quality of the isotopic measurements and the assessment of the error and potential bias in this study. However, I think that few sentences explaining why contamination by lithogenic material has the potential to bias the signal, and most importantly why this potential contamination is unlikely, or small, in this study is missing here. Is there any data from other studies that could support the fact that LSi is not a problem here?

**Results**
* Figure 2: Just an idea like this… adding a dark horizontal bar along the x axis to visualize the sampling period by the sediment trap.
* Figure 3: As I mentioned earlier, I am still not convinced that using the min/max this is the smartest way to present the isotope values, especially when refereeing only to the mean value in the text. I actually had to manually draw a line going through what should be the mean value for all the $\delta^{13}C$, $\delta^{15}N$ and $\delta^{30}Si$ panels to be able to properly follow authors' discussion (see picture below). I personally think that, as it is, the figure kind of work but does not easily help supporting the text and that it will greatly benefit by plotting one symbol for the mean value (as this is the one authors used in the discussion) and perhaps a vertical line representing the range between the min and max (if authors want to keep this information on the figure).
L361: "flux-weighted" sounds odd, perhaps use "integrated"
Table 1: Having an additional line with winter values will be useful as authors mention these winter values in the text.
L383: "[…] were globally/more or less similar […]"
L388-391: I don't see quite steady isotopic values in winter in the shallow trap. They vary from 1‰ to 1.5‰ which is a significant variation when considering $\delta^{30}Si$ values.
L410: Since Dictyocha is not a diatom, do we have an idea of the range of $\delta^{30}Si$ of these organisms and how they could potentially affect (or not) the isotopic signal measured in the traps?

**Discussion**
* L436-439: This could be more elaborated even briefly. For example, what kind of source? what are the different degradation states and how do they affect the $\delta^{15}N$ signal in the particles?
* L467: please change "algal" by "phytoplankton"
* L474-476: If it is only "broadly similar trends", and regarding the R2, I would not use "close coupling of carbon and silicon cycling processes."
* L508: Please remove "with no significant difference between deep and shallow" as deep trap data seem more variable compared to shallow trap data.
* L520: I would not qualify this as a slight increase (it just increases)
* L563-566: Could it be also associated with a shift in community with for example a little bit more silicoflagellates?
* L599 and few other times later in the discussion: "flux-weighted" sounds odd, perhaps use "integrated"

**Conclusion**
L704-706: Without changing the sentence too much, I think this study does not really highlight how, but perhaps more "the importance of conducting a more detailed mechanistic understanding of the drivers of POC flux […]"

NB: The data were not available at the time of the review

---

## Author Response (AR2)

**Response to reviewers: Seasonal cycles of biogeochemical fluxes in the Scotia Sea, Southern Ocean: A stable Isotope approach**

**Response to reviewers: Reviewer 1**

Review Biogeosciences Discussion (June 2023)/2nd round
Seasonal cycles of biogeochemical fluxes in the Scotia Sea, Southern Ocean: A stable
Isotope approach
Belcher et al.
The revised version is significantly improved. However, I still have some minor comments,
that needs to be addressed before publication.

Reply: Thank you for taking the time to review the manuscript, we have addressed the points you
have raised below. Line numbers refer to the marked up version of the text.

**L205**: The authors give the delta notation only for carbon and nitrogen isotopes, but not for
silicon isotopes. I see that the silicon method description is given in another paragraph, but
the authors could at least refer to the delta notation, as it is the same for all three stable
isotopes (C, N, Si).
Reply: We have added the following text to the methods (line 231 -234)
"Stable Si isotopic compositions are presented in standard delta notation ($\delta 30Si$), as for $\delta 13CPOC$
and $\delta 15NPN$ according to Equation 2, where R is 30Si/28Si. These compositions are checked against
$\delta 29Si$ (where R is 29Si/28Si) for mass dependence."

**L393:** Other organisms were observed, but not counted? Do the authors have a rough idea
of how much of the sediment trap material was diatoms compared to other (non-siliceous)
organisms)? This has some implications for the interpretation of $\delta 13C$ and $\delta 15N$. See also my
comment below (L.474). In line 395, the authors say "with a dominance of diatoms". What
exactly does that mean? More than 50% or 90%? The following paragraph only gives
information about the diatoms and a few silicoflagellates in each sample. How many
dinoflagellates do the authors observe? Does "other" in Figure 4 refers to other taxa, like
dinoflagellates? Or other diatoms? Please clarify in the text and the figure caption.
Reply: Other organisms were indeed counted, as detailed in the methods, but we focus the results
on the organisms that dominated the sediment trap material (as can be seen in figure 4). Figure 4
displays the phytoplankton that dominated by abundance as well as biovolume and it is clear from
this that diatoms are dominant. In most samples diatoms were >90% but all were greater than 85%
(by abundance and biovolume); we have added this into the text at line 411. The dinoflagellates
*prorocentrum* spp. and *pronoctiluca* spp. are shown to have >5% abundance in Figure 4, and the
silicoflagellate *Dictyocha* has >5% contribution by biovolume. 'Others' refers to all other taxa
counted to sum to 100%, so this encompasses any taxa that were counted but comprised <5% of the
sample. We have amended the caption to explain this.

**L429:** It is very interesting to see the comparison to other flux measurements in the region.
However, I miss some kind of interpretation here. Why are POC and BSi fluxes generally
higher, but much lower compared to Closset et al. (2015)? Any major changes in the area,
that are causing this. Why does the sampling location from Closset et al. (2015) have more
than 10x higher fluxes compared to this study?
Reply: There are a number of interacting factors impacting the magnitude of export flux, including
species composition, nutrient limitation and level of zooplankton grazing, and this, combined with
the high interannual variability in export flux in these high latitudes, makes it difficult to give the
exact cause of the differences between the two region. Comparing our study to that of Rembauville

highlights the high variability in the Scotia Sea, and where as Closset et al. 2015 measured very high fluxes, Trull et al. 2001, measured much lower fluxes in the same region in a different year.
We have added the following additional text lines 479-486.
"Interannual variability in export flux can be high due to the complexity of processes controlling the magnitude of export flux, such as community structure, nutrient limitation and zooplankton activity. Closset et al. (2015) measured very high fluxes (>700 mg $SiO_2$ $m^{-2}$ $d^{-1}$) of BSi south of the Sub-Antarctic Front in the Australian sector of the Southern Ocean at 2000 m, and similarly high fluxes have been observed in other sectors (Fischer et al., 2002; Honjo et al., 2000). A study by Trull et al. (2001) measured fluxes of BSi in the range of 30- 160 mg $SiO_2$ $m^{-2}$ $d^{-1}$ during the productive season in the same region as Closset et al. (2015), again highlighting the high interannual variability."

**L469:** Please check the sentence. Something is odd. "This could be achieved if cells are large through large".
Reply: We have added in a comma to correct this sentence

**L474:** I think, this is not even a "broadly" similar trend in Figure 5b. I think the authors should rather discuss, why they do not see a linear trend between δ13CPOC and δ30SiBSi. Even though the particulate ratios show a strong relation between POC and BSi (except for 3-4 points above the line), the less pronounced or not present relationship in the isotopes can have several reasons.
    1. more variation and a higher range in the fractionation factor for δ13C compared to δ30Si (e.g. Brandenburg et al., 20221), which can also include different trophic levels.
    2. non-siliceous organisms or organic material (dinoflagellates, microzooplankton). Whereas δ30Si is measured mainly in diatoms, δ13C, as well as δ15N, is measured in other materials/organisms as well
    3. different remineralization for organic carbon and silicon in the frustule

Reply: We have edited the text to reflect that though significant, the R2 is low. Thank you for the valuable insight here and suggestions of reasons to explain the weaker relationship between the Si and C isotopes. We have added this information in, as below, lines 513-519.

"Despite the strong relationship between particulate fluxes of POC and BSi, the relationship between the δ13CPOC and δ30SiBSi isotopes signatures is less pronounced (linear regression: $R^2 = 0.452$, $p<0.001$; Figure 5). This may relate to greater variation in the fractionation factor for δ13C compared to δ30Si (Brandenburg et al., 2022), as well as differences in remineralisation of organic carbon and silicon in the frustule. Additionally, whereas most of the δ30Si signal is from diatoms, the δ13C signal in the sediment trap material is also impacted by the presence of other organic material, e.g. zooplankton faecal pellets."

1 Brandenburg, K. M., Rost, B., Waal, D. B. V. de, Hoins, M. & Sluijs, A. Physiological control on carbon isotope fractionation in marine phytoplankton. *Biogeosciences* **19**, 3305–3315 (2022).

**L476:** The authors state that they do not find significant relationships between δ13CPON, δ13CPOC, and δ30SiBSi. It would be good if the authors could either show the figures in the supplement or report the r2 and p levels here for comparison. I am a bit surprised, that the relationship in Figure 5b is significant. Did you include the error? Please check again.
Reply: We have added in the p values of the non significant relationships as requested but do not give the R2 since these are 0 so do not provide useful information. We have double checked and the relationship in figure 5b is significant, we base the relationship on the mean values of Si and C isotope ratios from the replicated available.

**L487**: This is more of a general comment. Do the authors take the sinking velocity of particles into account, when discussing the sediment trap data? And if yes, what is the sinking velocity they assume?

Reply: We do not directly measure sinking rates since material can aggregate together in the sediment trap cup, and therefore particles cannot be presumed to be in the same state that that they were on entering the trap. We are able to make some inferences on sinking velocity based on lags between isotopic shifts in the shallow and deep sediment traps (e.g. see lines 343-349).

**L493**: "BSi: POC ratios were elevated at the start of productive period 1, suggesting that phytoplankton were heavily silicified…..this statement can only be made if the ratio of siliceous to non-non-siliceous plankton is not changing over time. Here the authors need to give more information about the amount of dinoflagellates in their samples (see also statement above). And if the statement is "true", why should a more intense silicification is observed?

Reply: Thank you for this observation, we have amended the text here as follows:

"BSi:POC ratios were elevated at the start of productive period 1, which may suggest that phytoplankton were heavily silicified. The contribution of non-siliceous phytoplankton was low during the periods analysed for phytoplankton composition (<2%, with the exception of the shallow trap in the late January sample where the contribution was 6.7%), though we cannot rule out higher contributions of non-siliceous phytoplankton during other periods which could account for the lower BSi:POC ratios at these times."

**Figure 3**: The figure did improve significantly. Maybe it is possible to additionally add the legend to the figure for the deep (red) and shallow (blue) sediment traps.

Reply: We have added the legend as below and amended the caption accordingly.

[Figure]

[Figure]

[Figure]

[Figure]

[Figure]

**Figure 4:** Maybe the authors could either highlight it in the figure or in the figure caption, how does the assemblage data fit to their different time periods? The abundance data at the end of the sampling campaign (Dec./Jan.) fit within the productive period 2, but the first was already in the winter hiatus.

Reply: We have added shading for the two productive periods as per figure 2

[Figure]

**Figure 5:** Please add error bars for the isotope data in 5b.

Reply: we have added error bars as per the error bars in figure 3, based on preservative uncertainty and analytical uncertainty for δ13CPOC and δ30SiBSi respectively.

[Figure]

**Response to reviewers: Reviewer 2**

**General comments**

In "Seasonal cycles of biogeochemical fluxes in the Scotia Sea, Southern Ocean: A stable isotope approach", Belcher et al. present a study investigating the seasonal variations of organic matter (POC and PON) and biogenic silica fluxes from two sediment traps located north-West of South Georgia in the Scotia Sea (Southern Ocean). Using stable isotope approaches the authors examine the origin and some of the processes controlling the fluxes they have observed in the traps.

They investigated the differences between two productive events (in February 2018 – summer season – and December 2018 – spring season) and the coupling of C, N and Si fluxes during these events. Their main results are: Particulate fluxes and isotopic compositions were similar in the deep and shallow trap suggesting that most of the remineralization occurred in the upper layer of the water column. Despite a very noisy d15N signal, the synchronicity if the d30Si and d13C signals highlight the coupling between these two elements and the significant role of diatoms in the export of C (and BSi) in the area. Based on the estimation of isotopic baselines associated with the two productive events, they also suggested a change in the source region of the material coming into the sediment traps.

Having reviewed the first version of this manuscript, I greatly appreciate authors' efforts to improve the reading by carefully re-organizing the different sections. The introduction is clear and describes all the background needed to fully appreciate the manuscript. New elements have been added to the discussion and greatly improve the manuscript. Some minor points will still benefit to be clarified and, although they have already greatly improved the figure, I am personally not convinced that figure 3, which is the most important figure in the manuscript, is not presented in the clearest/smartest possible way (but this is my personal taste, the data are currently there).

I detail these points below and, although I recommend publication of the manuscript after minor revisions, I am convinced that this paper will be a great addition to "Biogeosciences".

Thank you for taking the time to look at the manuscript for a second time and for the enthusiasm about the manuscript. Thank you for the minor points raised to further improve the manuscript, which we have addressed as detailed below. Line numbers refer to the marked up version of the text.

**Methods**

\* L238: Pioneer ref is Cardinal et al. 2003

Reply: Reference added in

\* L249-253: A 0.12‰ offset in δ30Si value might be an order of magnitude lower compared to the seasonal signal, however it is significant regarding the magnitude of δ30Si variations measured in the study (and could potentially be higher than the error calculated on duplicates or using standards). I am aware that this offset is the wort case scenario, and I am convinced about the quality of the isotopic measurements and the assessment of the error and potential bias in this study. However, I think that few sentences explaining why contamination by lithogenic material has the potential to bias the signal, and most importantly why this potential contamination is unlikely, or small, in this study is missing here. Is there any data from other studies that could support the fact that LSi is not a problem here?

Reply: The text here has been amended as follows:

"A lithogenic correction (e.g., Closset et al., 2015) was not carried out on these samples given the high percentage of biogenic silica present in the samples (mean percentage BSi as SiO2 of 17 %). BSi extraction methods show lower variability for marine sediments with BSi > 15-20 % and do not show evidence for significant leaching of lithogenic material through time (Conley, 1998). Furthermore, even an extreme scenario of variable lithogenic contamination of 1-5 % of isotopically light marine

clays (with δ30Si of -2.3 ‰; Opfergelt and Delmelle, 2012) would only result in a potential systematic offset of 0.12 ‰, which, although this is larger than the uncertainty on an individual datapoint, is an order of magnitude smaller than the observed seasonal signal."

**Results**
\* Figure 2: Just an idea like this… adding a dark horizontal bar along the x axis to visualize the sampling period by the sediment trap.
Reply: Thank you for the excellent suggestion, we have added this in and amended the caption accordingly.

[Figure]

\* Figure 3: As I mentioned earlier, I am still not convinced that using the min/max this is the smartest way to present the isotope values, especially when refereeing only to the mean value in the text. I actually had to manually draw a line going through what should be the mean value for all the δ13C, δ15N and δ30Si panels to be able to properly follow authors' discussion (see picture below). I personally think that, as it is, the figure kind of work but does not easily help supporting the text and that it will greatly benefit by plotting one symbol for the mean value (as this is the one authors used in the discussion) and perhaps a vertical line representing the range between the min and max (if authors want to keep this information on the figure).
Reply: We believe that figure 3 is now a good representation of the data, as agreed by the first reviewer. Since there are a few isotope values where the range is large between sample splits we think that showing the max and min more fairly represents the heterogeneity of sediment trap material. In these cases, the mean line (without max and min) is not a fair representation. We therefore keep figure 3 as is, adding the legend as requested by reviewer 1.

L361: "flux-weighted" sounds odd, perhaps use "integrated"
Reply: This is common terminology in sediment trap papers and is descriptive in terms of what has been used to weight the isotope values, so we keep this.

Table 1: Having an additional line with winter values will be useful as authors mention these winter values in the text.

Reply: We have added this in as suggested.

L383: "[…] were globally/more or less similar […]"
Reply: Amended to 'quite similar'

L388-391: I don't see quite steady isotopic values in winter in the shallow trap. They vary from 1‰ to 1.5‰ which is a significant variation when considering $\delta^{30}Si$ values.
Reply: We have amended the sentence to reflect the increase of 0.38 per mille in the shallow trap : "Isotopic values in the deep trap were then quite steady over winter compared to the rest of the record, with an increase of 0.38‰ in the shallow trap between May and August."

L410: Since Dictyocha is not a diatom, do we have an idea of the range of $\delta^{30}Si$ of these organisms and how they could potentially affect (or not) the isotopic signal measured in the traps?
Reply: To the best of our knowledge there have not yet been any studies investigating the range in $\delta^{30}Si$ of *Dictyocha* or indeed other silicoflagellates so we are not able to constrain the impact this of organism on our measured values. This would be a useful area of further research. We have added the following sentence to make it clear that we cannot assess the impact of this organism, though since the % contribution by abundance was <5% and diatoms were dominant (>85% - this information has been added to the manuscript), their isotopic signature would need to be vastly different from that of diatoms to have an appreciable impact on our results.
"Since there has been little known work on the $\delta^{30}Si$ of *Dictyocha* or indeed other silicoflagellates, we are not able to constrain the impact of this organism on our measured values. However, since the contribution by abundance was <5 % and diatoms were dominant (>85 %), their isotopic signature would need to be vastly different from that of diatoms to have an appreciable impact on our results."

**Discussion**
* L436-439: This could be more elaborated even briefly. For example, what kind of source? what are the different degradation states and how do they affect the $\delta^{15}N$ signal in the particles?
Reply:

Reply: We have expanded this argument, so that it is clear what we mean, as follows:

The lower POC:PN ratios measured in the deep trap between August and October, compared to the shallow trap are consistent with a divergence in $\delta^{15}N_{PN}$ ratios, and could indicate that material arriving at the two traps is not necessarily sourced from the same region and time period in surface waters. Given the slower sinking speeds at this low-productivity time of year, it is possible that material reaching the deep trap is sourced from upstream of where material reaching the shallow trap is sourced in the regional circulation system. Different source regions are likely characterised by different phytoplankton assemblages with different nutrient stoichiometry, and the time taken for source material to reach each of the traps may well lead to differences in degradation state of organic matter, which could also lead to variations in POC:PN.

To ensure that the reviewer's comment is addressed comprehensively, we have also added to the related argument in Section 4.2.2 (lines 591-602), as follows.

"$\delta^{15}N_{PN}$ in the shallow trap showed a slight progressive decrease from April to July, before increasing in August to 5.42 ‰. The progressive decrease is consistent with the propagation of the surface signal of phytoplankton growth and fractionation, with a longer time lag than during spring and summer due to slower sinking rates during the low-productivity period. Decreasing $\delta^{15}N_{PN}$ reflects the increasing influence of ammonium uptake, either in the same locale or upstream in the regional

circulation system, which leads to lower $\delta^{15}N_{PN}$ than nitrate uptake in the slowly-sinking flux. The large range in $\delta^{15}N_{PN}$ in the deep trap in July makes it difficult to determine with certainty a trend in $\delta^{15}N_{PN}$ in the deep trap between July and October. Dissimilar trends in $\delta^{15}N_{PN}$ between the two traps over the winter period also support the argument that material reaching these two traps may have a different source region or time period in surface waters (Section 4.1)."

* L467: please change "algal" by "phytoplankton"
Reply: Changed

* L474-476: If it is only "broadly similar trends", and regarding the R2, I would not use "close coupling of carbon and silicon cycling processes."
Reply: We have amended this section in response to this comment and a related comment from reviewer 1, as follows:
"Despite the strong relationship between particulate fluxes of POC and BSi, the relationship between the $\delta^{13}C_{POC}$ and $\delta^{30}Si_{BSi}$ isotope signatures is less pronounced (linear regression: $R^2 = 0.452$, p<0.001; Figure 5). This may relate to greater variation in the fractionation factor for $\delta^{13}C$ compared to $\delta^{30}Si$ (Brandenburg et al., 2022), as well as differences in remineralisation of organic carbon and silicon in the frustule. Additionally, whereas most of the $\delta^{30}Si$ signal is from diatoms, the $\delta^{13}C$ signal in the sediment trap material is also impacted by the presence of other organic material, e.g. zooplankton faecal pellets."

* L508: Please remove "with no significant difference between deep and shallow" as deep trap data seem more variable compared to shallow trap data.
Reply: Deleted

* L520: I would not qualify this as a slight increase (it just increases)
Reply: Deleted word slight

* L563-566: Could it be also associated with a shift in community with for example a little bit more silicoflagellates?
Reply: Although species related shifts in silicon isotopes have been observed in the laboratory, the evidence for this is lacking in the field (e.g.Closset et al., 2015; Egan et al., 2012), and thus we do not speculate about this here. We mention this in lines 587-591 and 697-699, and do not think it needs repeating here. We have no constraints on Si isotope fractionation by silicoflagellates.

* L599 and few other times later in the discussion: "flux-weighted" sounds odd, perhaps use "integrated"
Reply: As above, this is common terminology in sediment trap papers and is descriptive in terms of what has been used to weight the isotope values, so we keep this.

**Conclusion**
L704-706: Without changing the sentence too much, I think this study does not really highlight how, but perhaps more "the importance of conducting a more detailed mechanistic understanding of the drivers of POC flux […]"
Reply: Amended as follows:
"Our results highlight the need for more detailed mechanistic understanding of the drivers of POC flux and biogeochemical cycling, to improve estimates of the current and future strength of the biological carbon pump and the ocean's role as a $CO_2$ sink."

NB: The data were not available at the time of the review

Reply: The data DOI are in progress and data will be housed with the Polar Data Centre, as per the data availability statement.